# How Well Do CMIP6 Models Simulate the Greening of the Tibetan Plateau?

Jiafeng Liu [1,2] and Yaqiong Lu [1,*]

1 Institute of Mountain Hazards and Environment, Chinese Academy of Sciences, Chengdu 610299, China
2 University of Chinese Academy of Sciences, Beijing 100049, China
* Correspondence: yaqiong@imde.ac.cn; Tel.: +86-02861158015

**Abstract:** The "warm-humid" climate change across the Tibetan Plateau (TP) has promoted grassland growth and an overall greening trend has been observed by remote sensing products. Many of the current generations of Earth System Models (ESMs) incorporate advanced process-based vegetation growth in the land surface module that can simulate vegetation growth, but the evaluation of their performance has not received much attention, especially over hot spots where projections of the future climate and vegetation growth are greatly needed. In this study, we compare the leaf area index (LAI) simulations of 35 ESMs that participated in CMIP6 to a remote-sensing-derived LAI product (GLASS LAI). The results show that about 40% of the models overestimated the Tibetan Plateau's greening, 48% of the models underestimated the greening, and 11% of the models showed a declining LAI trend. The CMIP6 models generally produced poor simulations of the spatial distribution of LAI trend, and overestimated the LAI trend of alpine vegetation, grassland, and forest, but underestimated meadow and shrub. Compared with other vegetation types, simulations of the forest LAI trend were the worst, the declining trend in forest pixels on the TP was generally underestimated, and the greening of the meadow was underestimated as well. However, the greening of the grassland, was greatly overestimated. For the Tibetan Plateau's averaged LAI, more than 70% of the models overestimated this during the growing seasons of 1981–2014. Similar to the forest LAI trend, the performance of the forest LAI simulation was the worst among the different vegetation types, and the forest LAI was underestimated as well.

**Keywords:** Coupled Model Intercomparison Project Phase 6 (CMIP6); LAI; LAI trend; Tibetan Plateau

## 1. Introduction

Vegetation is a critical component of terrestrial ecosystems and is very sensitive to climate change [1–3]. The global average surface temperature increased by 0.85 °C from 1880 to 2012 [4], which triggered phenological changes in different vegetation types in different regions. The increase in temperature, as one of the causes of variation in vegetation, has led to a significant overall change in vegetation, manifested by an increase in the Normalized Difference Vegetation Index (NDVI) during the vegetation growth season in the Northern Hemisphere [5], and the growth rate of NDVI in forests is greater than that of other vegetation types [6–8]. The community structure of snow-meadow vegetation has changed significantly as a result of climate change in Northern Japan over the last 40 years [9]. In the Siberian Mountains, the birch area has increased by 10%, and birch stands and the treeline boundary have moved upslope at a rate of 1.4 m yr⁻¹ and 4.0 m yr⁻¹, respectively, since the 1970s with the onset of warming [10]. In China, the zone of tundra vegetation of the Changbai Mountains has been invaded by herbaceous plants with the rising temperature over the last 30 years [11].

As the third pole of the earth, the Tibetan Plateau (TP) is highly sensitive to climate change and has been experiencing a rapid warming of 0.4° 10 yr⁻¹ over the last 30 years [12,13] and with precipitation increasing by 1.96 mm 10 yr⁻¹ in 1994–2015 [14]. This "warm–humid" trend has led to tremendous changes on the land surface, such as glaciers collapsing [15], permafrost thawing [16], and lakes expanding [17], as well as surface vegetation growth. Liu et al. [18] found that the vegetation coverage on the TP showed a trend of "overall increase and partial degradation" from 1981 to 2005, with the area of improvement much larger than the area of degradation. Wei et al. [19] found that "warm-humid" has a significant promoting effect on the improvement of vegetation on the TP, and Zhang et al. [20] found that the overall NDVI of grassland in the growing season of the TP also shows an increasing trend. Xu et al. [21] used the leaf area index inversion by NOAA–AVHRR to study the temporal and spatial changes in vegetation cover characteristics in the TP, and also found an overall increase in vegetation cover. Zhang et al. [13] found that the green-up dates with the alpine vegetation in the Plateau had a continuous advancing trend with a rate of ~1.04 d·y⁻¹ from 1982 to 2011.

Remote sensing, as one of the major tools for studying vegetation's response to climate change [22], was used to study the vegetation on the TP, with various long-term vegetation leaf area index (LAI) datasets derived through satellite remote sensing, such as GLASS LAI [23], GLOBMAP LAI [24], GIMMS LAI [25], and MODIS LAI [5]. Hua et al. (2018) [26] used the GIMMS NDVI dataset (NDVI-3g) to study the temporal and spatial variations in vegetation dynamics controlled by climate on the Tibetan Plateau during 1982–2011 and found that the potential cause of the change in vegetation dynamics might be controlled by the climate, particularly the increasing precipitation and the significant temperature rise in the Central and Southeastern Tibetan Plateau. Although remote sensing products are very useful for understanding historical vegetation variations, satellite remote sensing could not directly measure future vegetation dynamics. Another powerful tool, the state-of-the-art Earth System Models that incorporate a process-based vegetation growth module, can simulate not only historical variations in vegetation but also those in future climate. Zhu et al. [27] built the first pedotransfer function to simulate temporal variations in vegetation coverage (VC) and found that the pedotransfer function more accurately simulated temporal variation in VC than a multiple linear regression in an alpine meadow on the Tibetan Plateau. Lu et al. [28] found that net primary productivity (NPP) and LAI decreased from the southeast to the northwest of the Tibetan Plateau by using the atmosphere–vegetation interaction model (AVIM) to simulate the distribution of LAI and NPP over the Tibetan Plateau. The accuracy of the simulation results varies greatly due to the design and use of the model itself, so it is very important to evaluate the accuracy of the simulation data before using the simulations.

The International Coupled Model Comparison Program (CMIP), proposed by the World Climate Research Program Group, currently in the sixth generation (CMIP6), has been widely used for studying various environmental changes. Tian et al. [29] analyzed changes in the annual mean surface air temperature (SAT) and precipitation, and also the related uncertainties using historical simulations and future projections under the Representative Concentration Pathway scenarios (RCPs) from the CMIP5 models across China and in its seven sub-regions. Zhang et al. [30] demonstrated that there may be a basic spatial scale limit below which it may not be useful to further refine climate model predictions based on an integrated analysis of coupled model simulations and projections from CMIP3 and CMIP5. Using the established linear relationship and monthly temperature simulations from CMIP5 models over the Northern Hemisphere during the 21st century, Xia et al. [31] found the start of the vegetation growing season (SOS) will have advanced by 4.7 days under RCP2.6 (Representative Concentration Pathway) by 2040–2059. After CMIP5, more and more models have incorporated a dynamic vegetation growth module, and therefore evaluating CMIP vegetation simulations has drawn much attention. Anav et al. [32] assessed the ability of 18 Earth system models (ESMs) in CMIP5 and found that most models overestimated the global average LAI and half of the models also

overestimated the LAI trend for 1986–2005. Zhao et al. [33] analyzed the changes in projected global LAI from 16 CMIP5 ESMs and 17 CMIP6 ESMs, and found that the CMIP6 models had a better ability to describe the global area-averaged LAI time series. Lawrence et al. [34] did not evaluate the performance of the simulated global tree height of the CMIPs' ESMs but gave the biases of tree height for the offline simulations of CLM5BGC. Brovkin et al. [35] evaluated the performance of MPI-ESM, and Seller [36] evaluated UKESM1-0-LI in terms of vegetation distribution; both found that the two models overestimated the fraction of tree coverage. Most evaluations have focused on the global scale; few have focused on regional scales such as the Tibetan Plateau. Bao et al. [37] evaluated 12 CMIP5 ESMs for reproducing vegetation cover and LAI over the Tibetan Plateau in 1986–2005, and found that INMCM4, BCC-CSM-1.1M, MPI-ESM-LR, IPSL-CM5A-LR, HadGEM2-ES, and CCSM4 were the best six models for capturing vegetation among the 12 models. CMIP6 has had the largest participation since its implementation [38]. However, how well the CMIP6 models simulate vegetation growth, especially the recent greening of the Tibetan Plateau, is unknown.

LAI is usually defined as half of the total leaf surface area per unit of surface area [39], and NDVI is defined as the ratio of the difference between the near-infrared band (NIR) and the visible red band (R), and the sum of the two bands, NDVI = (NIR-R)/(NIR+R). NDVI is directly obtained from the satellites' reflection information and the real-time variation of vegetation after a simple calculation, which can quantitatively reflect the actual variation of vegetation, including the vegetation structure, the vegetation growth, and the vegetation coverage during the observation period, and is widely used in the field of vegetation remote sensing [40–42]. LAI and NDVI are both important indices for quantifying the vegetation variations, but only LAI could be validated because NDVI is not an output of the dynamic vegetation growth models in CMIP6. LAI, as a key indicator of vegetation growth [43], has been widely used in global climate models, ecological models, hydrological models, and ecosystem productivity models [44]. Therefore, we focused on LAI validations in our work rather than NDVI.

In recent decades, although greening is one of the most important changes in the Tibetan Plateau, few works have particularly focused on the performance of the model simulations on the greening of the Tibetan Plateau. We developed our own ranking method that considered the temporal and spatial simulations' abilities to give an overall assessment of CMIP6 models. We also quantified the growth of different vegetation types. Our goals with this work are to evaluate the performance to simulate the LAI trend and LAI of the CMIP6 model during the growing season and to provide a reference for the selection of simulation data of vegetation changes, aid the research into vegetation in the Tibetan Plateau, and analyze the sources of temporal and spatial error in each model, laying a foundation for model optimization.

## 2. Data and Methods

### 2.1. Study Area

The TP [45,46] is located at 26–39°N latitude in Southwest China. Surrounded by high mountains on the edge of the area, the internal topography is complex, including plateaus, basins, glaciers, lakes, and swamps [47]. Its geographical features, such as the high altitude, and the complex and changeable topography, have created special climatic conditions and water and heat distribution in this area, and have also created its distinctive vegetation distribution. As the largest alpine grassland ecosystem in the world, the TP is dominated by meadows and grasslands (Figure 1), concentrated across a wide range of Central Tibet. The vegetation types in Tibet have spatial distribution characteristics that gradually change from southeast to northwest. From southeast to northwest in Tibet, the vegetation types are distributed in the order of forests, shrubs, meadows, grassland, and desert (Figure 1). The dataset is derived from the 1:1 million vegetation data set collected

in China in 2001, and it is provided by the National Cryosphere Desert Data Center (http://www.ncdc.ac.cn) (accessed on 9 December 2021).

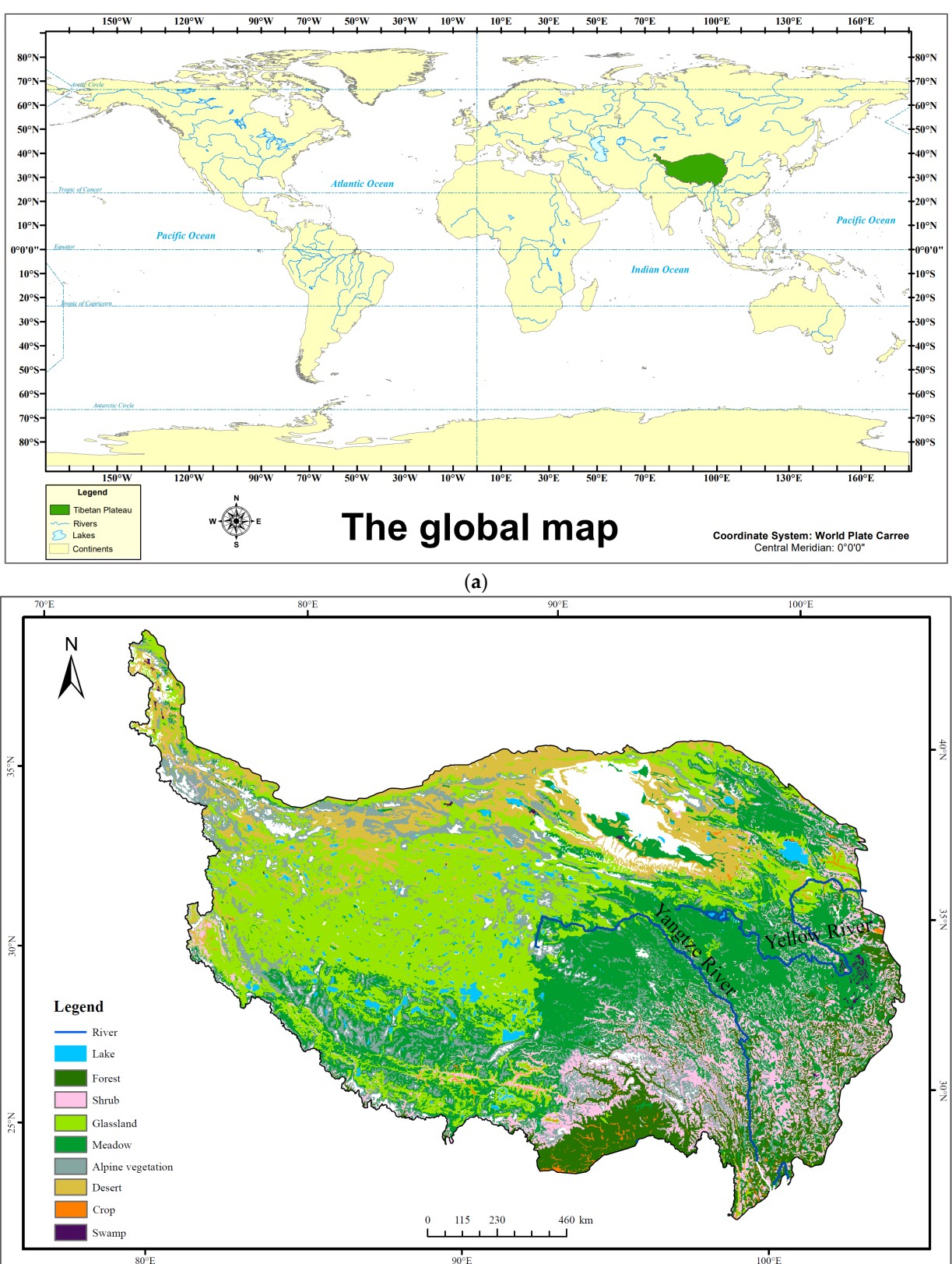

**Figure 1.** (**a**) The location of the Tibetan Plateau [48] in the world map, and the world map from ArcGIS. (**b**) Distribution of vegetation types on the Tibetan Plateau [49].

*2.2. Satellite Data*

To evaluate the ability of the 35 models from the CMIP6 to reproduce the LAI over the Tibetan Plateau, the 1981–2018 LAI data from the Global Land Surface Satellite (GLASS) dataset with an eight-day temporal frequency and a 0.5° × 0.5° spatial resolution were used as a benchmark in our study. GLASS LAI uses generalized regression neural networks (GRNNs) to invert LAI from Time-Series AVHRR Surface Reflectance data; the algorithm trains GRNNs using preprocessed AVHRR Time-Series AVHRR Surface Reflectance, and then uses rolling processing to produce time-continuous long-term GLASS LAI products from the preprocessed AVHRR Surface Reflectance [23]. Compared with other LAI datasets, GLASS LAI data have a long observation period, high quality, and good accuracy [50]. They have more complete trajectories than the MODIS LAI product and also show lower uncertainty than the MODIS and CYCLOPES LAI products compared with 20 ground-measured LAI reference maps. Many studies use GLASS LAI as a reference database for research or validation [51–55]. All these factors make it an ideal long-term dynamic LAI observation dataset in this study. The GLASS LAI product (V50) used in this study is available from the University of Maryland and the Center for Global Change Data Processing and Analysis of Beijing Normal University (http://www.glass.umd.edu/Download.html, accessed on 9 March 2021).

*2.3. CMIP6 Model Simulations*

Thirty-five CMIP6 models with no missing data were selected in this study, and the LAI from outputs of historical simulations for 1850–2014 was used (https://esgf-node.llnl.gov/search/cmip6/, accessed on 16 August 2021).

In order to facilitate the comparison of the simulation and observational data, all simulations were downloaded and converted to a 0.5° × 0.5° spatial resolution by bilinear interpolation from low to high resolution. The overlaps of the GLASS datasets and CMIP6 were 1981–2014, so our analysis focused on 1981–2014. The model's information is shown in Table 1.

**Table 1.** Model description.

| Model | Institute | Land Surface Model | Resolution | Reference |
|---|---|---|---|---|
| AWI-ESM-1-1-LR | AWI (Germany) | CABLE2.4 | 250 km | [56] |
| ACCESS-ESM1-5 | CSIRO (Australia) | CABLE2.4 | 250 km | [57] |
| BCC-CSM2-MR | BCC (China) | BCC-AVIM2.0 | 100 km | [58] |
| BCC-ESM1 | BCC (China) | BCC-AVIM2.0 | 250 km | [58] |
| CAMS-CSM1-0 | China | CoLM | 100 km | [59] |
| CanESM5 | CCCMA (Canada) | CLASS3.6-CTEM1.2 | 500 km | [60] |
| CanESM5-CanOE | CCCMA (Canada) | CLASS3.6-CTEM1.2 | 500 km | [60] |
| CESM2 | NCAR (USA) | CLM5 | 100 km | [61] |
| CESM2-FV2 | NCAR (USA) | CLM5 | 100 km | [61] |
| CMCC-CM2-SR5 | CMCC (Italy) | CLM4.5 | 100 km | [62] |
| CMCC-ESM2 | CMCC (Italy) | CLM4.5 | 100 km | [62] |
| E3SM-1-0 | E3SM-Project (USA) | ELM | 100 km | [63] |
| E3SM-1-1 | E3SM-Project (USA) | ELM | 100 km | [63] |
| E3SM-1-1-ECA | E3SM-Project (USA) | ELM | 100 km | [63] |
| EC-Earth3-Veg | EC-Earth-Consortium (Europe) | HTESSEL | 100 km | [64] |
| EC-Earth3-Veg-LR | EC-Earth-Consortium (Europe) | HTESSEL | 100 km | [64] |
| FGOALS-g3 | China | CAS-LSM | 2 × 2° | [65] |
| FIO-ESM-2-0 | FIO (China) | CLM4.0 | 100 km | [66] |
| GFDL-CM4 | GFDL (USA) | LM4.0 | 100 km | [67] |
| GFDL-ESM4 | GFDL (USA) | LM4.1 | 100 km | [68] |
| GISS-E2-1-G | GISS (USA) | GISS LSM | 250 km | [69] |

| HadGEM3-GC31-LL | HadGEM (United Kingdom) | JULES | 250 km | [70] |
|---|---|---|---|---|
| HadGEM3-GC31-MM | HadGEM (United Kingdom) | JULES | 100 km | [70] |
| INM-CM4-8 | INM (Russia) | INM-LND1 | 100 km | [71] |
| INM-CM5-0 | INM (Russia) | INM-LND1 | 100 km | [72] |
| IPSL-CM6A-LR | IPSL (France) | ORCHIDEE v2 | 250 km | [73] |
| KIOST-ESM | KIOST (Korea) | LM3.0 | 250 km | [74] |
| MIROC-ES2L | MIROC (Japan) | MATSIRO6.0 +VISIT-e v1 | 500 km | [75] |
| MPI-ESM-1-2-HAM | HAMMOZ Consortium (Switzerland, Germany, Finland, UK) | CABLE2.4 | 250 km | [76] |
| MPI-ESM1-2-HR | MPI (Germany) | CABLE2.4 | 100 km | [76] |
| MRI-ESM2-0 | MRI (Japan) | HAL 1.0 | 100 km | [77] |
| NorESM2-LM | NCC (Norway) | CLM5 | 250 km | [78] |
| NorESM2-MM | NCC (Norway) | CLM5 | 100 km | [78] |
| TaiESM1 | AS-RCEC (Taiwan, China) | CLM4.0 | 100 km | [79] |
| UKESM1-0-LI | MOHC (UK) | JULES-ES-1.0 | 250 km | [36] |

*2.4. Evaluation Approach*

A series of evaluation indicators was applied to quantify the agreement between the observed and simulated LAI and the trend of the CMIP6 models. In this study, we calculated the average LAI during the growing season (May–September) for each year as the average LAI, a linear regression trend of the average LAI from 1981 to 2014 as the trend, and an increasing trend indicated TP greening. We also calculated the monthly average LAI for each month of the growing season, and the TP averaged monthly average LAI during 1981–2014 as the monthly LAI. Then, we calculated the linear regression trend of the monthly average LAI for each month during the growing season from 1981 to 2014, and the TP averaged trend of the monthly average LAI as the monthly LAI trend. We obtained monthly variations from the monthly LAI and the monthly LAI trend during the growing season. In the following, we further describe the metrics used for model evaluation and the method used for ranking the models.

2.4.1. Evaluation Metrics

The spatial correlation (pattern correlation) was used to quantify the correlation between the grid cell trend (or the grid cell average LAI from 1981 to 2014) distribution in the models and observations. Through a combination of the definitions of Bao et al. [37] and Chang et al. [80], the spatial correlation formula for the simulated and observed trends in this study was defined as follows:

$$Pattern\ correlation = \frac{\frac{1}{N}\sum_{i}^{N} W_i \left(M_i - \overline{M}\right)\left(O_i - \overline{O}\right)}{\sqrt{\frac{1}{N}\sum_{i}^{N} W_i \left(M_i - \overline{M}\right)^2}\sqrt{\frac{1}{N}\sum_{i}^{N} W_i \left(O_i - \overline{O}\right)^2}} \quad (1)$$

where $N$ is the total number of grid cells under evaluation, $M_i$ and $O_i$ are the simulated and observed trend (or the average LAI from 1981 to 2014) from the CMIP6 models and the GLASS of the grid cell $i$, and $W_i$ is the area weight of the grid cell $i$ (all grid weights add up to 1) [37]. We calculated $W_i$ in the Pearson correlation coefficient equation as the area of each grid cell associated with the central geographic latitude of each grid cell [37]. In the TP, the variation in $W_i$ is not obvious and the value of $W_i$ can almost be neglected.

The bias between the simulated and observed grid cell trend (or the grid cell average LAI from 1981 to 2014) was calculated to quantify the main bias between the model simulations and GLASS observations. In our study, we subtracted the observed trend (or the average LAI from 1981 to 2014) from the simulated trend (or the average LAI from 1981

to 2014) to get trend (or the average LAI from 1981 to 2014) bias at the single grid cell $i$ by Equation (2). We thus obtained a value of the bias at every grid cell and the distribution of the bias across the whole study region. The relative bias of grid cell trend (or the grid cell average LAI from 1981 to 2014) was calculated as the ratio of the trend (or the average LAI from 1981 to 2014) bias to the observed trend (or the average LAI from 1981 to 2014) at the grid cell $i$ in Equation (3). We also calculated the TP averaged bias using Equation (4).

$$Bias = M_i - O_i \tag{2}$$

$$Relative_{Bias} = \frac{Bias_i}{O_i} \tag{3}$$

$$Bias_{avg} = \frac{\sum_1^N |M_i - O_i|}{N} \tag{4}$$

The root-mean-square error (RMSE) was used to measure the difference between the simulations and observations. Similar to bias, we calculated the trend (the average LAI from 1981 to 2014) of the two datasets at grid cell $i$, and then aggregated the results over the entire TP. Next, we converted spatial two-dimensional data of trend (or the average LAI from 1981 to 2014) in simulations and observations into one dimension and calculated the RMSE of the two columns (the simulations and observations) of the one-dimensional data by Equation (5). This RMSE was used in the ranking. Moreover, we used RMSE to quantify the difference in the average LAI from 1981 to 2014 sequence between the simulation and observation during the growing season in 1981–2014 at single grid cell $i$, and then obtained the distribution of RMSE across the region.

$$RMSE = \sqrt{\frac{\sum_1^N \left( M_i - O_i \right)^2}{N}} \tag{5}$$

The ratio of the standard deviation ($Ratio_\delta$) was used to quantify the magnitude of the difference in variation between the simulation and the observation. Similar to RMSE, we first converted the spatial two-dimensional data of the grid cell trend (or the grid cell average LAI from 1981 to 2014) in simulations and observations into one dimension, and then calculated the standard deviation of the simulations and observations by Equation (6), and finally calculated the ratio of the two standard deviations. Furthermore, $\delta_M$ and $\delta_O$ were the standard deviations of the model simulations and the GLASS observations, respectively. The ratio of trend ($Ratio_{trend}$) was used to quantify the variation of the simulated trend and the observed trend as either overestimation or underestimation. We calculated the $Ratio_{trend}$ by Equation (7); $trend_M$ was the simulated trend, and the $trend_O$ was the GLASS trend. A ratio less than 0 indicated that the trend was not captured, contrary to the trend in GLASS. A ratio greater than 0 but less than 1 indicated that the greening or declined trend was captured, but was underestimated. A ratio greater than 1 indicated an overestimation of the greening or declined trend.

$$Ratio_\delta = \frac{\delta_M}{\delta_O} \tag{6}$$

$$Ratio_{trend} = \frac{trend_M}{trend_O} \tag{7}$$

### 2.4.2. Significant Test Method

We used two methods for significance testing, the Student's *t*-test and the Mann–Kendall trend test. The Student's *t*-test was used for the significant difference test between simulations and observations. The Mann–Kendall trend test was used to detect whether a time series was steadily increasing/decreasing or unchanging.

### 2.4.3. Ranking Method

A ranking scheme was developed by Brunke et al. to score the multi-bulk aerodynamic algorithm for calculating the turbulence fluxes on the ocean surface [81]. Decker et al. [82] ranked the bias and standard deviation of error between reanalysis products and flux tower measurements using the same method as Brunke et al. On the basis of Decker et al., Wang et al. [83] extended this ranking approach and increased the statistical parameters to four, including the correlation coefficient ($\varrho$), the standard deviation ratio ($\sigma_r/\sigma_{obs}$), the standard deviation error ($\sigma_d$), and the difference (bias) to rank the ability of six kinds of reanalysis data to reproduce climate characteristics over the Tibetan Plateau. Since then, this ranking approach, as a good example of model performance evaluation, has been used in many studies [32,37]. In this study, we adjusted the ranking method used by Wang and Zeng, and the ranking metrics were changed into the spatial correlation (pattern correlation), the bias (Bias), the root mean square error (RMSE), and the ratio of standard deviation ($Ratio_\delta$).

In the context of this study, the simulation with the highest pattern correlation, the lowest bias and RMSE, and the closest ratio was considered to have the best performance for reproducing the trend (or the LAI) over the Tibetan Plateau. The models were ranked from 1 to 35, with 1 being the model with the lowest value in magnitude of bias, RMSE, or |ratio-1| (or the highest pattern correlation) and 35 being the model with the highest value in magnitude of bias, RMSE, or |ratio-1| (or the lowest pattern correlation) [82]. We then calculated the total score of the four metrics for a single model and defined the total score as the "error ranking". The higher the model's error ranking, the closer the relationship between the simulations and observations.

## 3. Result

### 3.1. The Average Growing Season LAI and Trend

More than 70% of models overestimated and about 28% of models underestimated the area-averaged growing season LAI over the Tibetan Plateau (Figure 2). EC-Earth3-Veg, C-Earth3-Veg-LR, and HadGEM3-GC31-LL showed the smallest average LAI bias with slight underestimations of 0.0066–0.018 $m^2$ $m^{-2}$ in comparison with GLASS LAI. CMIP6 models (except FI0-ESM-2-0) incorporating the community land model (hereafter referred to as the CLM family) showed a much larger LAI bias of 2–5.5 $m^2$ $m^{-2}$, especially CESM2, CESM2-FV2, NorESM2-LM, and NorESM2-MM (4–5.5 $m^2$ $m^{-2}$). CanESM5, CanESM5-CanOE, E3SM-1-0, GISS-E2-1-G, IPSL-CM6A-LR, and KIOST-ESM underestimated the average LAI (0.1–0.40 $m^2$ $m^{-2}$), but these underestimations were much smaller than the overestimations of other CMIP6 models.

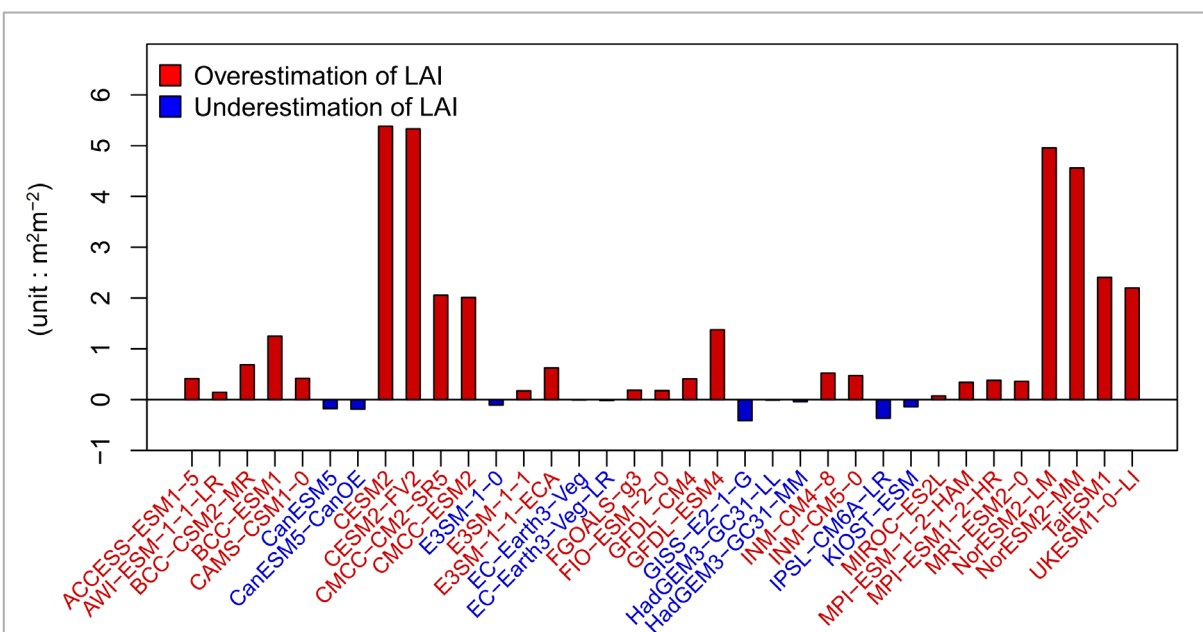

**Figure 2.** The bias of the area-averaged LAI during the growing season in Tibetan Plateau from 1981 to 2014 between each CMIP6 model and GLASS data.

In Figure 3, we show the ratio of the area-averaged trend between simulations and observations from 1981–2014 in TP. For the Tibetan Plateau LAI trend in 1981–2014, about 40% of the models overestimated the Tibetan Plateau's greening, more than 48% of the models underestimated the greening, and 11% models showed a declining LAI trend (Figure 3). E3SM-1-1 and MPI-ESM-1-2-HAM showed the closest trend estimations among the 35 CMIP6 models. For some CMIP6 models, the overestimation or the underestimation of greening and the area-averaged LAI (in Figure 2) occurred at the same time. For example, CMIP6 models (except for FI0-ESM-2-0) that incorporated CLM also greatly overestimated the greening of the Tibetan Plateau above the GLASS data (2.5–6.5 times higher), while CanESM5 underestimated not only the average LAI but also the greening. However, models such as AWI-ESM-1-1-LR and UKESM1-0-LI overestimated the average LAI but underestimated the greening.

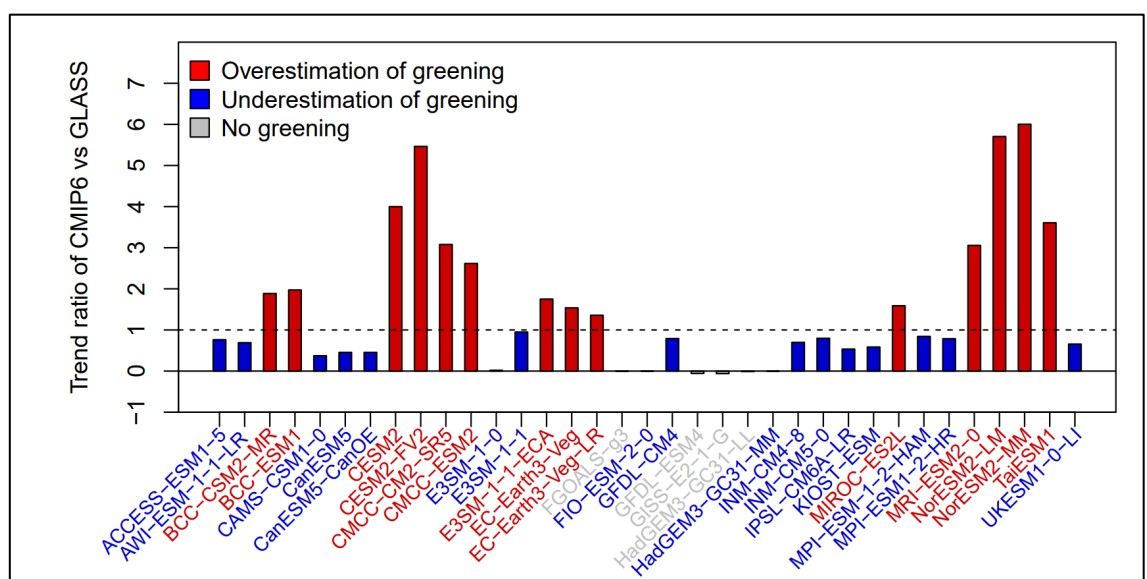

**Figure 3.** The ratio of the area-averaged LAI trend of the growing season (1981–2014) between each CMIP6 models and the GLASS data.

### 3.2. LAI and Trend Monthly Variations

3.2.1. Monthly Leaf Area Index

The maximum underestimation of LAI mainly occurred in July and August, while the maximum overestimation of LAI varied greatly across different CMIP6 models, and this variation depended greatly on the land surface models incorporated in the different CMIP6 models (Figure 4a). The monthly variation in the bias of the LM family (UKESM1-0-LI, GFDL-CM4, and GFDL-ESM4) was similar for each month of the growing season. Unlike the LM family, the overestimation bias of the CLM family (except for FIO-ESM-2-0) first increased and then remained stable, with the bias in May being the smallest, and the largest being in June or September. The bias of the BCC family showed more complex monthly variation characteristics, with the overestimation bias increasing and then decreasing, and the bias in August being the largest.

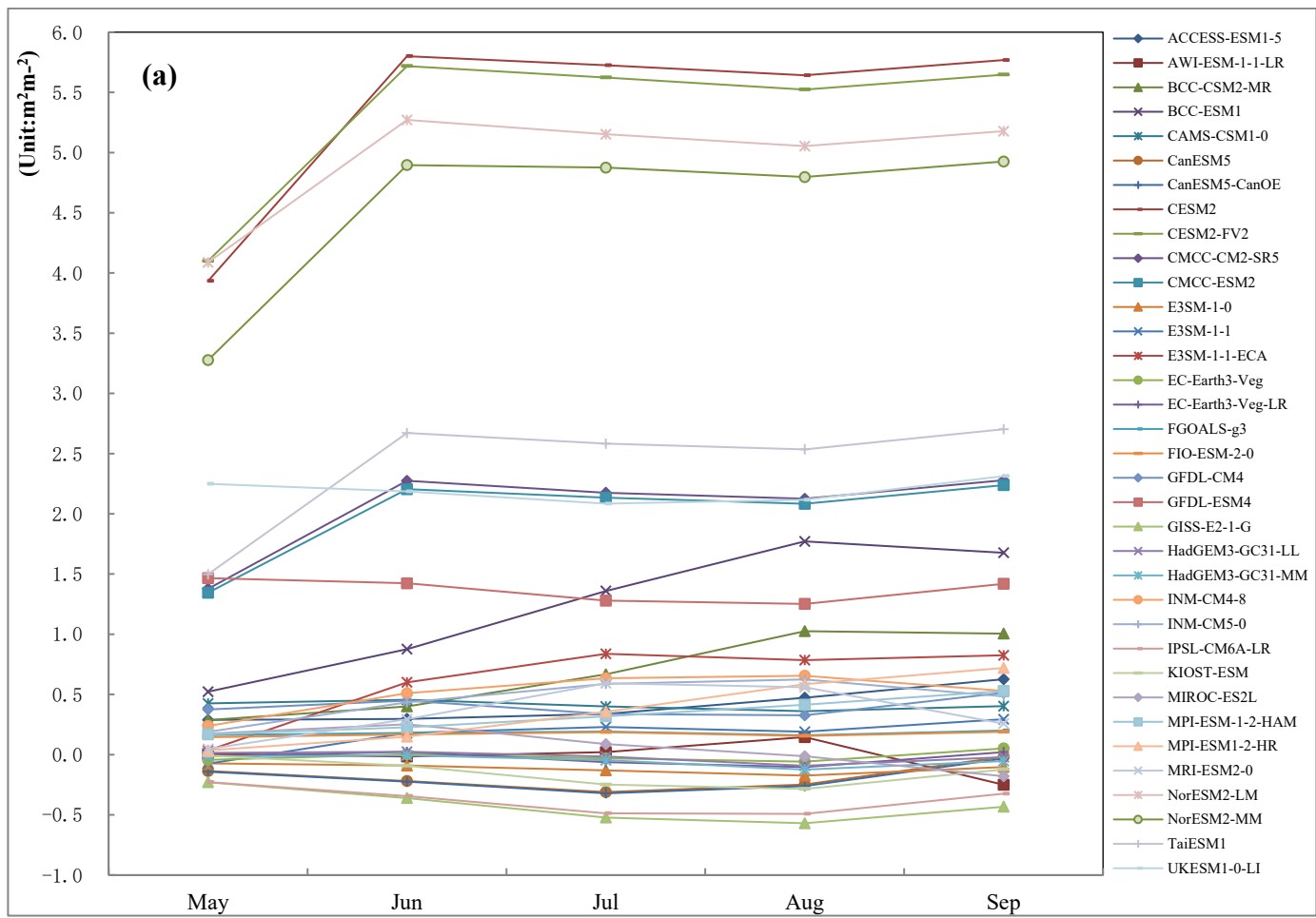

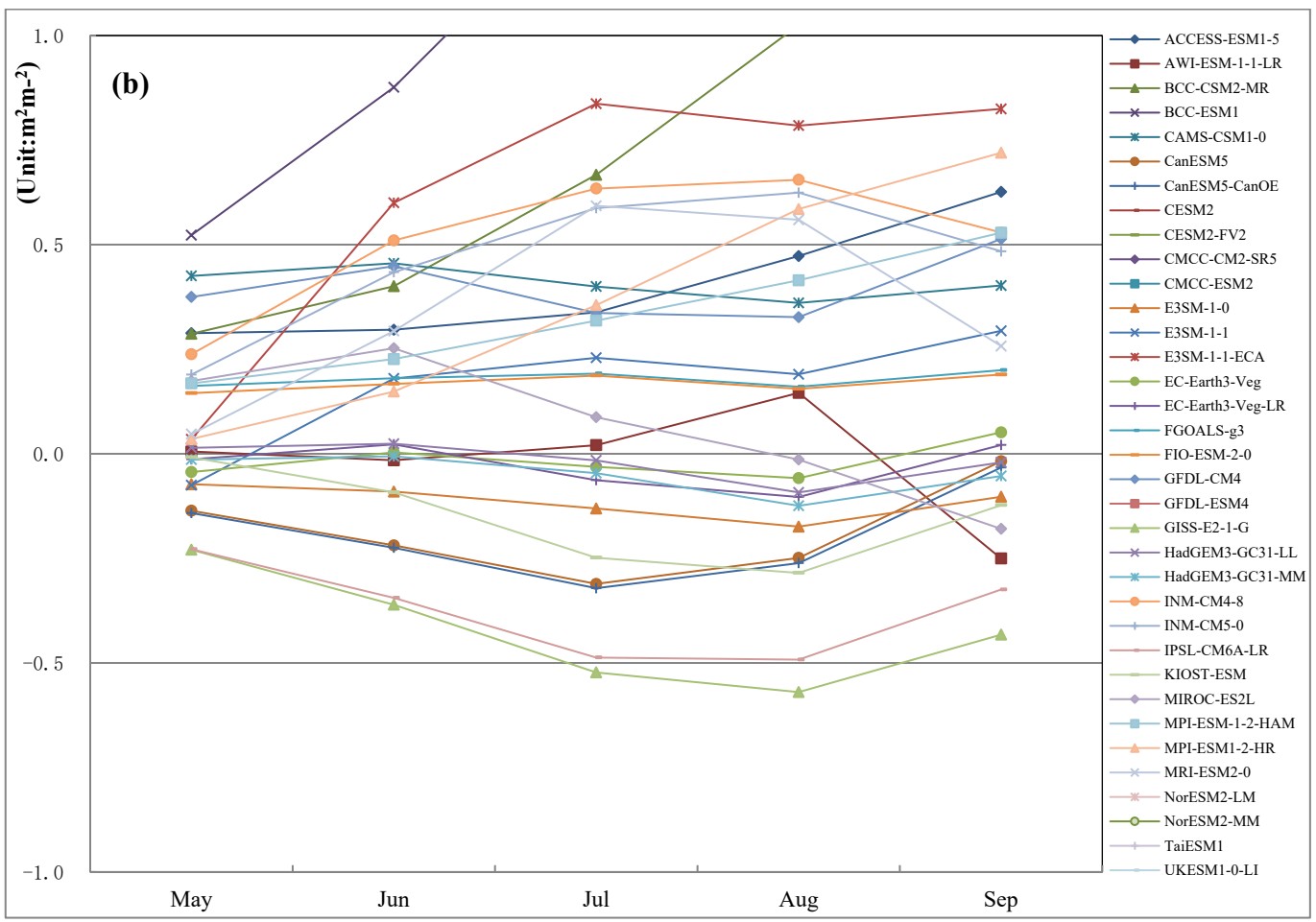

**Figure 4.** The bias of the monthly LAI during the growing season of the Tibetan Plateau between each CMIP6 model and the GLASS data. The *y*-axis is from −1 to 6 in (**a**), and from −1 to 1 in (**b**).

Moreover, the good simulations of area-averaged LAI of EC-Earth3-Veg, EC-Earth3-Veg-LR, and HadGEM3-GC31-LL were due to the positive and negative biases in different months cancelling each other out. EC-Earth3-Veg and EC-Earth3-Veg-LR underestimated LAI in May (−0.04 to −0.01 m² m⁻²), July (−0.06 to −0.03 m² m⁻²), and August (−0.1 to −0.05 m²m⁻²), while LAI was overestimated in June (0.003–0.022 m² m⁻²) and September (0.02–0.05 m² m⁻²), HadGEM3-GC31-LL overestimated LAI in May (0.014 m² m⁻²) and June (0.024 m² m⁻²), but underestimated LAI in July (−0.016 m² m⁻²), August (−0.09 m² m⁻²), and September (−0.021 m² m⁻²), and these biases partially canceled each other out, making the overall average bias smaller.

Although the bias of LAI in May was small, the relative LAI bias was quite large in May (Figure S1). For example, the relative LAI bias of the CLM family (except for FIO-ESM-2-0) was highest in May and June (364–1105%) and then decreased from May or June to August (265–725%), which suggested that improvements at the beginning of growth are key to these models.

### 3.2.2. Monthly LAI Trend

None of the CMIP6 models captured the monthly LAI trend well, even those models that showed good agreement for the annual LAI trend (Figure 5). The good overall greening simulations of E3SM-1-1, INM-CM5-0, INM-CM4-8, and MPI-ESM1-2-HR were due to the overestimations and underestimations in different months cancelling each other out.

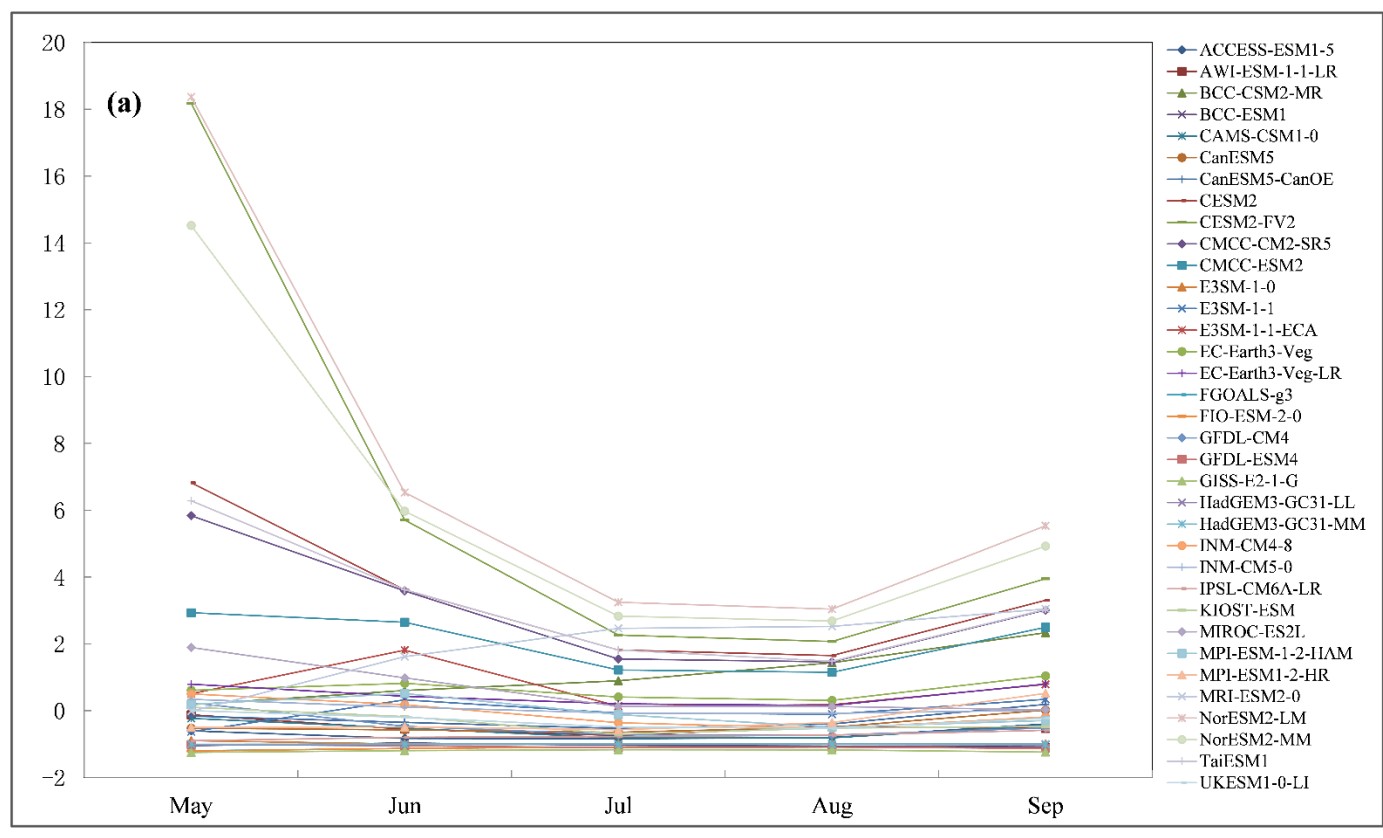

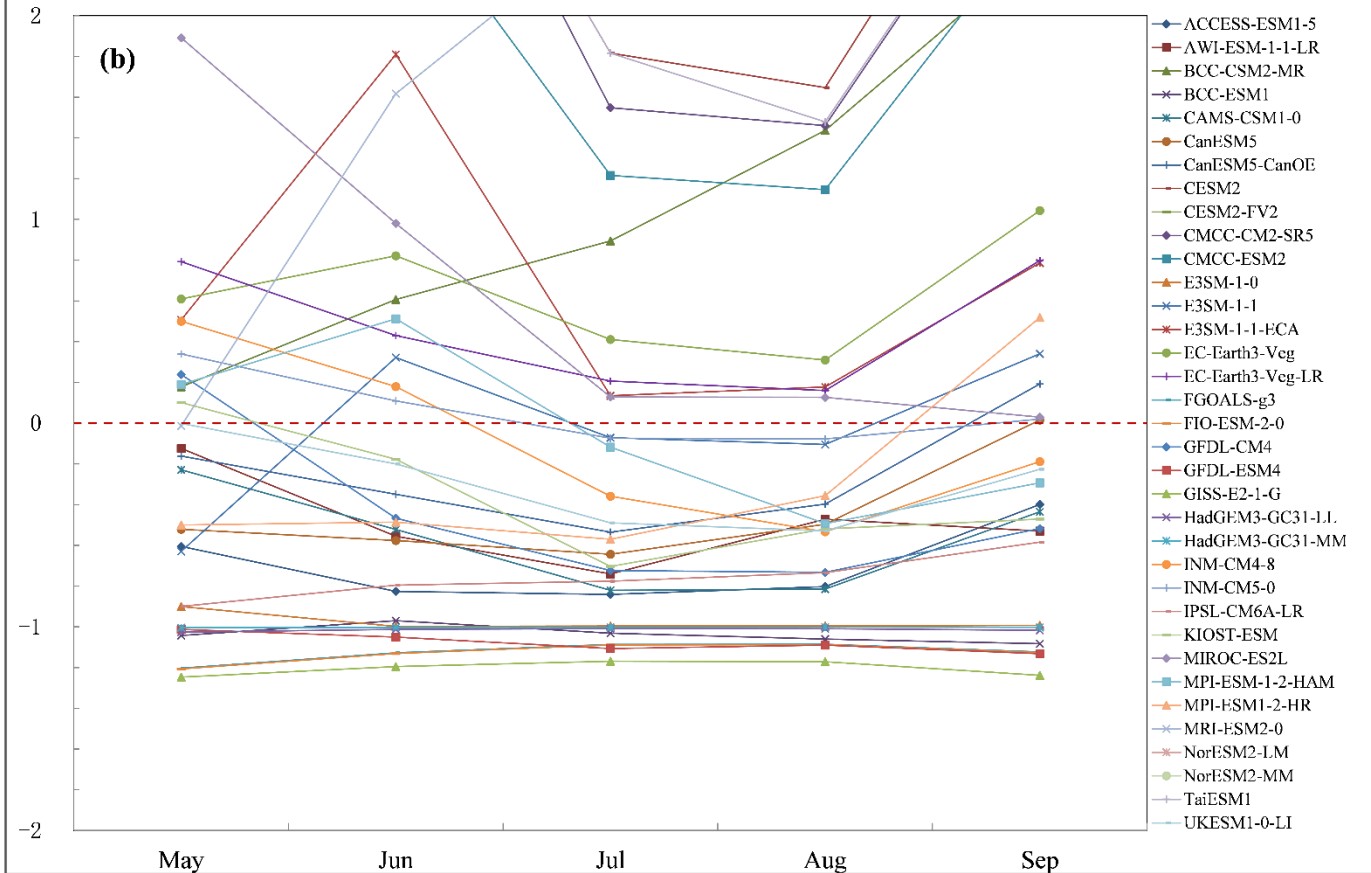

**Figure 5.** The ratio of the monthly LAI trend of the growing season between each CMIP6 model and the GLASS data. The *y*-axis is from −2 to 20 in (**a**), and from −2 to 2 in (**b**).

Models that underestimated the greening of the Tibetan Plateau generally had the greatest underestimation in July and August (Figure 5). For example, except for IPSL-CM6A-LR, the monthly error of other models that underestimated the greening of the Tibetan Plateau showed the changes first increasing and then decreasing, and the underestimation error was usually the largest in July and August. However, the models that overestimated the greening of the Tibetan Plateau showed inconsistent monthly variations. For example, the CLM family (except for FIO-ESM-2-0) showed the largest overestimation in May (2.93–18.37) and the smallest overestimations in July (1.22–3.24) and August (1.15–3.04). BCC-CSM2-MR showed the greatest overestimation in September (3.33), while E3SM-1-1-ECA showed the greatest overestimation in June (2.81). The models that did not simulate greening also did not simulate the greening trend for each month of the growing season.

Unlike the large difference between the LAI bias and relative LAI bias, the ratio of the monthly LAI trend and the bias of the monthly LAI trend had consistent variations (Figure S2). The CLM family (except for FIO-ESM-2-0) showed the largest overestimation in May, and the greatest underestimation of LAI trend in July and August.

### 3.3. LAI Spatial Comparison

3.3.1. Averaged Leaf Area Index for 1981–2014

GLASS LAI gradually decreased from southeast to northwest (Figure 6). The LAI of forests in Southeast TP was larger (2.8–4.8 $m^2$ $m^{-2}$), and the LAI dominated by grasslands and shrubs in the central and northwest areas was smaller (0–0.8 $m^2$ $m^{-2}$).

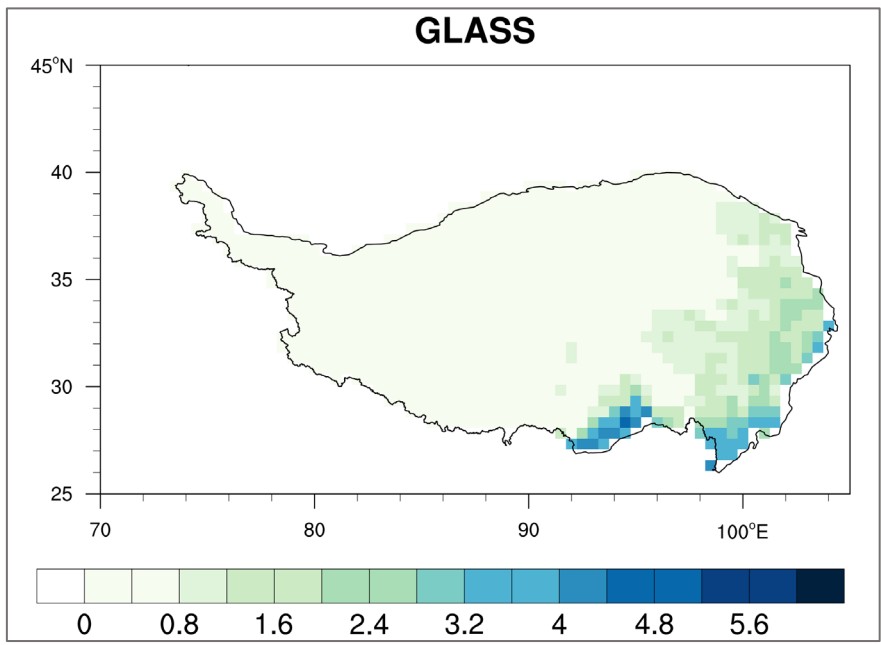

**Figure 6.** Spatial distribution of the GLASS LAI during the growing season.

Before evaluating the spatial distribution simulation capability, we ranked the performance of the CMIP6 models to capture the LAI spatial distribution based on the evaluation metrics (Table S1), then we presented the LAI spatial distribution results in Figures S3 and 7 by ranking their scores from the best to the worst.

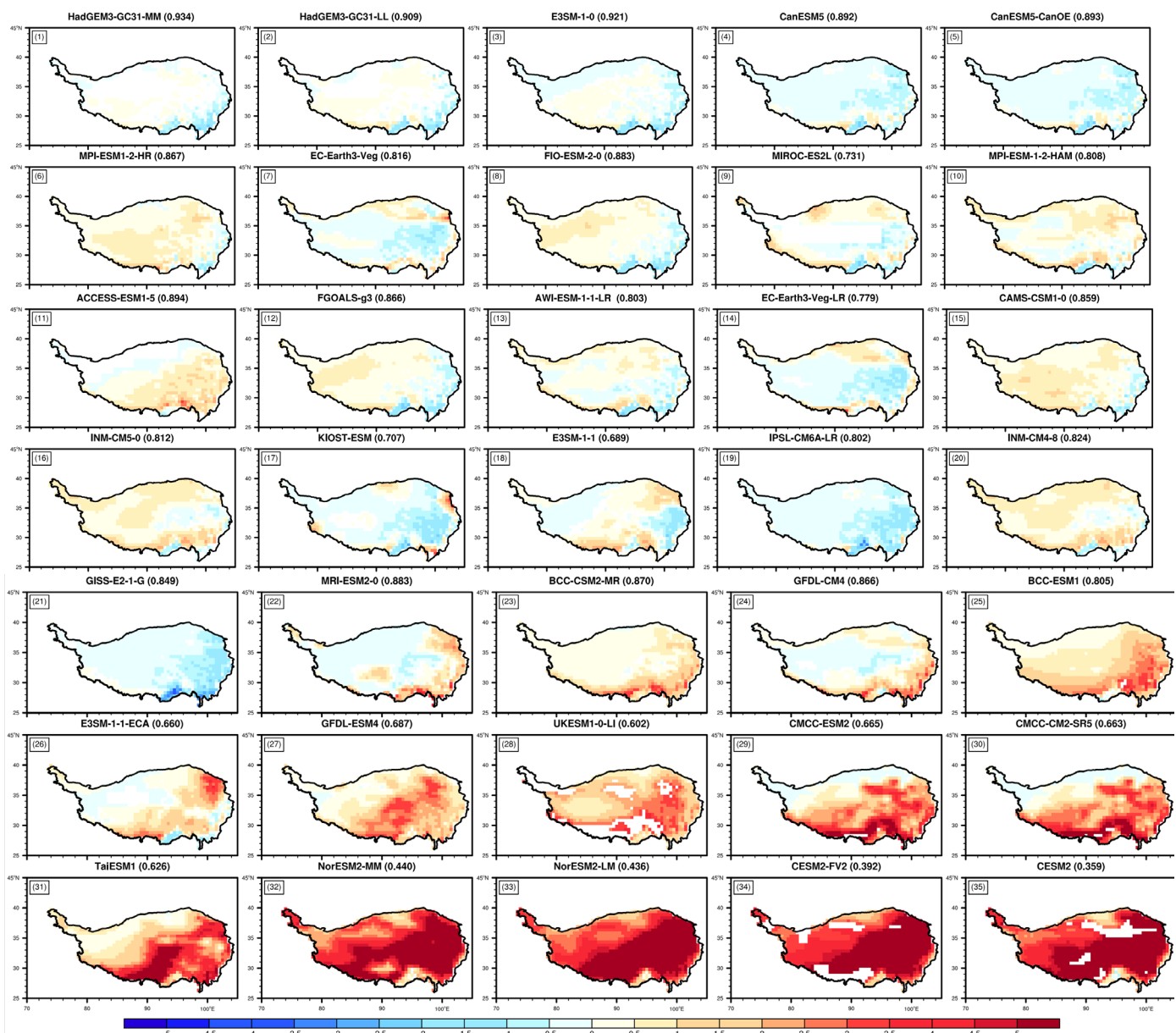

**Figure 7.** Spatial distribution of the bias of simulated and observed LAI during the growing season. The white part failed the significant difference test. The number in the top left corner is the ranking of each CMIP6 model for simulating the spatial distribution of the average LAI during the growing season in 1981–2014. The value in each title is the pattern correlation.

Almost all the CMIP6 models could reproduce a spatially declining pattern from southeast to northwest, but there was still large spatial bias. The pattern correlation of 88% of the models was greater than 0.60 and the highest was 0.934 for HadGEM3-GC31-MM (Figure S3). We also found that the top five models among the 35 CMIP6 models mainly underestimated the LAI, and the underestimation bias mainly came from the alpine forest area and alpine meadow areas in southeast Tibet. The main feature of the model ranked in the middle (ranked 6–20) among the CMIP6 models is that there were both overestimations and underestimations in the region, while the models with lower (after 20) rankings mainly overestimated the LAI, and the overestimation bias was more obvious in the southeast.

Models that underestimated LAI did so mainly over meadows and alpine forest areas in southeast TP, while models that obviously overestimated LAI had great differences in their spatial bias (Figure 7). The obvious overestimation of BCC-CSM2-MR from the BCC

family mainly came from the shrub area, while the overestimation of BCC-ESM1 was mainly from shrub areas, meadows, and part of the grassland, and there was high over-estimation near river basins. The overestimation of GFDL-CM4 in the LM family came from the shrub areas, while the overestimation of GFDL-ESM4 was mainly distributed across all of Southeast Tibet and was significantly overestimated in the central part. The overestimation of the INM family mainly occurred in shrub areas, deserts, and grassland area, and the highest value of the overestimation bias was for the shrub areas. In addition, the CLM family (except for FIO-ESM-2-0) had abnormally high LAI values throughout the Tibetan Plateau region, and the overestimation was distributed throughout the region, especially for shrub areas and meadows in the southeast TP, with the bias values being 4.5–5.0 m² m⁻².

EC-Earth3-Veg and EC-Earth3-Veg-LR showed the best simulations for the average LAI (Figure 2), but none of them showed the exact spatial distribution of LAI (Figure 7). EC-Earth3-Veg and EC-Earth3-Veg-LR overestimated the southeastern edge of Tibet but underestimated the grassland and meadow regions of the TP; these positive and negative errors cancelled each other out.

Overall, the CMIP6 models had poor performance for the forest LAI simulation with the highest RMSE, and the bias of the CMIP6 models varied greatly with large overesti-mation and underestimation, but with the smallest relative bias (Figure S4). Although CMIP6 models had a small overestimation of forest average LAI generally (Figure S4), most models underestimated the forest LAI in the small areas where forests are concentrated on the southern edge of the TP (Figure 7). Similar to the forest LAI, the simulation of shrub was poor with large RMSE and bias, but the relative bias of the shrub was small. The perfor-mance of the CMIP6 models for simulating the grassland LAI was good among the different vegetation types with the smallest RMSE. The reason for the small absolute bias but large relative bias with grassland may be that the LAI value of grassland was small.

### 3.3.2. The Leaf Area Index Trend during 1981–2014

The GLASS LAI data showed a clear greening trend from 1981 to 2014 over the TP, except for some forest areas on the southern edge of the TP (Figure 8). The entire area had significant greening ($p < 0.05$) of 0.0047 m² m⁻² yr⁻¹ (Figure S4), especially in the river basins of the meadow area.

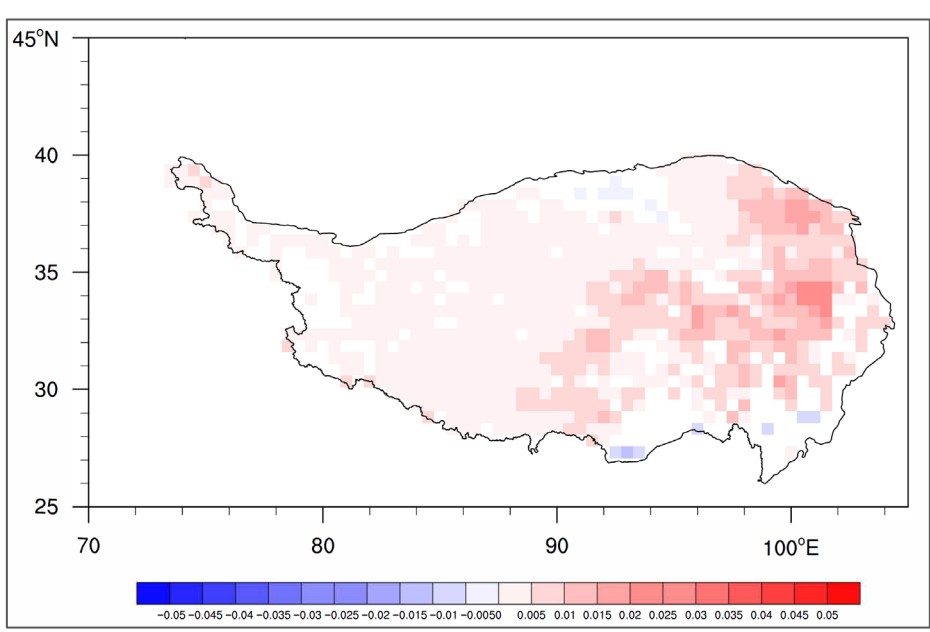

**Figure 8.** Spatial distributions of the linear trend of GLASS LAI during the growing season.

Similar to the analysis of the spatial distribution of the LAI, we ranked the performance in reproducing the LAI trend of the CMIP6 models (Table S2) and show the spatial distribution of the LAI trend from best to worst in Figure 9.

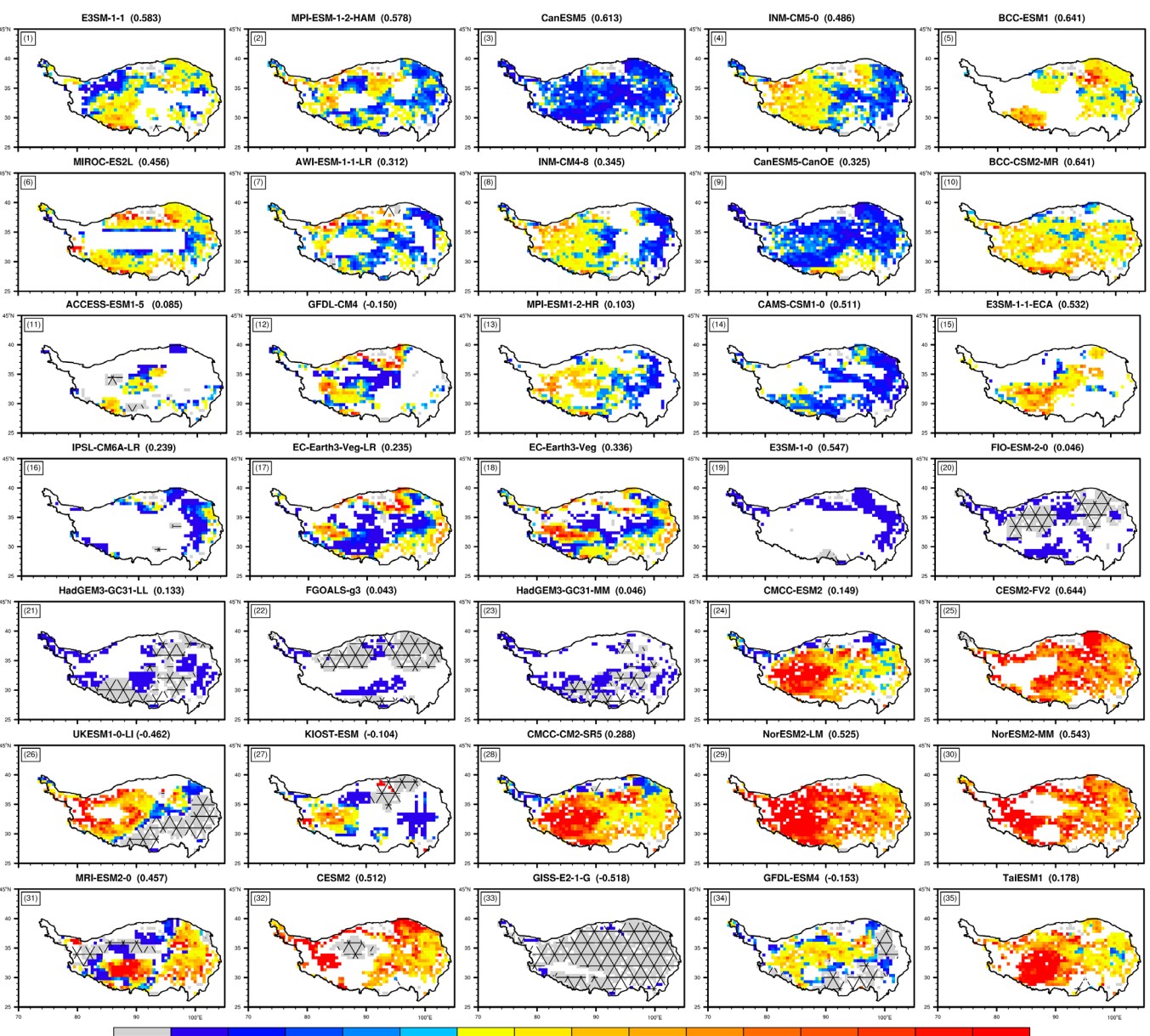

**Figure 9.** Spatial distributions of the ratio of simulated and observed linear trends in LAI during the growing season. The grid cells with colors all showed a statistically significant interannual change ($p < 0.05$). Gray areas mean the grid cells did not capture greening or a declining trend during 1981–2014 in the Tibetan Plateau, blue areas mean the grid cells captured the greening or the declining trend but underestimated them, and red areas indicated overestimations of the greening or the declining trend. Cross-hatched areas indicate that the LAI trend was negative. The number in the upper left corner is the ranking of each CMIP6 model for simulating LAI trends. The value in each title is the pattern correlation.

The CMIP6 models showed a poor ability to simulate the spatial distribution of the LAI trend across the whole Tibetan Plateau during 1981–2014, while most models could simulate the LAI trend in parts of the Tibetan Plateau (Figure 9). The pattern correlation of the LAI trend between all models and GLASS was less than 0.65, and a few models even had negative pattern correlations (Figure S6). There were five models (MPI-ESM-1-2-HAM, BCC-ESM1, BCC-CSM2-MR, EC-Earth-Veg, and EC-Earth-Veg-LR) that

generally simulated the overall greening trend of the study area, and also captured the high value of the greening trend in the southeast region, where the spatial distribution of the greening trend was closer to the observation data, and five models (E3SM-1-1, AWI-ESM-1-1-LR, CESM2, GFDL-ESM4, and TaiESM1) simulated the obvious decline in the Southern TP better than other 30 models. E3SM-1-1 and MPI-ESM-1-2-HAM had the best performance in simulating the distribution of the LAI trend and could capture the greening and the decline as well.

Compared with other vegetation types, the simulation of forest LAI trend was poor with the highest RMSE, and the CMIP6 models generally overestimated the forest LAI trend. The simulation of the forest LAI trend showed great differences. Some models showed largely overestimations (NorESM2-MM with a bias of 0.026 $m^2$ $m^{-2}$ $year^{-1}$) and some models showed large underestimations (GFDL-ESM4 with a bias of –0.017 $m^2$ $m^{-2}$ $year^{-1}$), which resulted in a larger LAI bias range across all CMIP6 models than for other vegetation types (Figure S7). The alpine vegetation and grassland were also overestimated by CMIP6 models, but the meadow and shrub were underestimated (Figure S7).

In total, 70% of the models accurately simulated increases and decreases in the LAI trend of 80% of the area of the Tibetan Plateau, but the simulation of the value of the LAI trends on the grids was poor (Figure 9). Six models (FIO-ESM-2-0, HadGEM3-GC31-LL, FGOALS-g3, UKESM1-0-LI, GISS-E2-1-G, and GFDL-ESM4) all had obvious gray areas, which mean that the models showed a contrary trend to the GLASS data and had not captured the greening or the declining—especially for GISS-E2-1-G, the gray area was distributed across almost the entire area. Neither FIO-ESM-2-0 nor FGOALS-g3 captured the LAI trend in Northern Tibet, and neither UKESM1-0-LI nor GFDL-ESM4 captured the LAI trend in the southwestern region.

The remaining models all captured the greening in 1981–2014, while there were still underestimations and overestimations of the value of the LAI trend in grid cells (Figure 9). The underestimation of the LAI trend mainly came from the shrub, whole meadow area or part of the meadow area, and the greening of the shrub and meadows was underestimated. While the overestimation of the LAI trend came from the grasslands, the CLM family (except for FIO-ESM-2-0) overestimated the LAI trend in almost the whole area, especially the greening of the grassland, which was greatly overestimated. Similarly, 13 models (E3SM-1-1, INM-CM5-0, MIROC-ES2L, INM-CM4-8, MRI-ESM2-0, GFDL-CM4, MPI-ESM1-2-HR, E3SM-1-1-ECA, EC-Earth-Veg, EC-Earth-Veg-LR, UKESM1-0-LI, KIOST-ESM, and MRI-ESM2-0) all overestimated the greening of grasslands. Although the trend of forest LAI was generally overestimated by CMIP6 models (Figure S7), the decline trend of forest LAI was underestimated in parts of the southeast where alpine forests were concentrated (Figure 9).

## 4. Discussion

Our study chose GLASS LAI as our reference LAI because it is one of the leading data sources for studying long-term series vegetation changes with good representations of various surface LAI distributions. In an evaluation of the authenticity of GLASS LAI products in the grasslands of Xilinhot [84], it was found that the observational accuracy and consistency of GLASS LAI were better than those of MODIS LAI, making it more suitable for related research. When GLASS LAI data were used to analyze changes in the Amazon rainforest from 1982 to 2012 [53], it was demonstrated that the GLASS LAI data can be used for detecting changes in the large-scale surface vegetation status in long sequences. As early as 2014, Xiang et al. [85] compared LAI products (MODIS LAI, CYCLOPES LAI, and CCRS LAI) with ground measurement LAI data, and found that the accuracy of GLASS LAI data products was significantly higher than that of MODIS and CYCLOPES. At the same time, through a comparison of LAI products (MODIS LAI, CYCLOPES LAI, and CCRS LAI), it was found that, compared with other LAI products, GLASS LAI has the best temporal continuity and integrity, and smoother trajectories, and is an ideal data product for studying temporal changes in LAI. The spatial distribution of the GLASS LAI

data is reasonable, and it also has good consistency with the global spatial distribution of MODIS LAI. It thus has great advantages in studies of the spatial distribution of LAI.

Although many studies using remote sensing products found an overall increasing trend of vegetation growth (greening) over the Tibetan Plateau, like the GLASS remote sensing products, controversy remains regarding how vegetation on the Tibetan Plateau has changed. Xu showed that spring warming advanced spring leaf-out time and increased the biomass [86]. However, Yu (2010) argued that the warm winter may also have led to delayed spring phases due to insufficient fulfillment of chilling requirements [87]. Zhang et al. [13] argued for the earlier start date of plant phenology and a longer growing season, but some still doubt this [88,89]. Regarding the change in the trend, a declined trend for the vegetation dynamics of the TP was found in some studies over the last 30 years (about 1980-2010) according to the Global Inventory of Modeling and Mapping and Studies (GIMMS) [90,91], but others found an increasing trend of vegetation growth in northeastern TP using other NDVI datasets for 1982–2011 [92]. These different understandings indicate that a combination of ground observations, remote sensing datasets, and land/vegetation models is necessary to fully understand past and future vegetation changes on the Tibetan Plateau.

In the CMIP6 models, some model groups showed consistency in simulating LAI and LAI trends using the same land surface model, but others showed great differences. In order to understand the possible reasons for these differences, we briefly summarized the differences among the model groups using the same land surface model (Table 2).

**Table 2.** Summary of the different models.

| Land Surface Model | CMIP6 Models | The Difference of Models |
| --- | --- | --- |
| BCC-AVIM2.0 | BCC-CSM2-MR<br><br>BCC-ESM1 | BCC-CSM2-MR uses the carbon emissions provided by CMIP6 as the forcing, but BCC-ESM1.0 uses the chemical reaction gas and aerosol emission data provided by CMIP6 as the forcing [93] |
| CLASS3.6-CTEM1.2 | CanESM5<br>CanESM5-CanOE | CanESM5-CanOE is exactly the same physical model as CanESM5, but it couples it with the CanOE ocean biogeochemical model [60] |
| CLM4.0 | FIO-ESM-2-0<br><br>TaiESM1 | The FIO-ESM-2-0 model adds an ocean surface wave model to the traditional atmosphere–land–ocean–sea ice coupled model of CPL7; TaiESM1 was developed on the basis of the Community Earth System Model version 1.2.2 by implementing several improvements to the parameterization schemes in the atmospheric component [94,95] |
| CLM4.5 | CMCC-CM2-SR5<br><br>CMCC-ESM2 | CMCC-CM2-SR5 does not include ocean biogeochemistry model in the model, but the BFM5.1 ocean biogeochemistry model was added to CMCC-ESM2 |
| CLM5.0 | CESM2<br>CESM2-FV2<br>NorESM2-LM<br>NorESM2-MM | CESM2-FV2 reduces the horizontal resolution of the atmosphere and land on the basis of CESM2; NorESM2-LM and NorESM2-MM are similar to the CESM2 and CESM2-FV2 models in terms of the framework and model composition; the differences are that NorESM2 uses completely different oceans and oceano-biogeochemical model and uses a different ocean and oceano-biogeochemistry model and the atmosphere component of NorESM2-MM and CAM-Nor; the difference between NorESM2-LM and NorESM2-MM is the resolution [78,96] |
| ELM | E3SM-1-0<br>E3SM-1-1<br>E3SM-1-1-ECA | On the basis of E3SM-1-0, E3SM-1-1 has corrected several vulnerabilities and made improvements; on the basis of E3SM-1-1, E3SM-1-1-ECA uses the ECA plant and soil carbon and nutrient mechanisms, soil carbon and the effects of nutrients representing carbon, nitrogen and phosphorus, and it excludes the effect of coupled ocean and sea ice biogeochemistry [97] |
| HTESSEL | EC-Earth3-Veg | EC-Earth3-Veg-LR has a lower resolution than EC-Earth3-Veg |

| | EC-Earth3-Veg-LR | |
|---|---|---|
| INM-LND1 | INM-CM4-8<br>INM-CM5-0 | On the basis of INM-CM4-8, the key improvements in INM-CM5-0 include an increase in the vertical resolution in the atmospheric module, a revision of the large-scale condensation and cloud formation parameterizations, the newly developed aerosol block, the horizontal resolution of the oceanic model, and a reworking of the INMCM5 program code for better performance on parallel computers [71] |
| JSBACH 3.2 | AWI-ESM-1-1-LR<br>MPI-ESM-1-2-HAM<br>MPI-ESM1-2-HR | AWI-ESM-1-1-LR is based on AWI-ESM and adds a dynamic land change model to it; MPI-ESM1-2-HR and MPI-ESM-1-2-HAM both are based on MPIESM1.2, and the difference between the two is that MPI-ESM-1-2-HAM adds the Hamburg aerosol mode and MPI-ESM1-2-HR improves the resolution of MPIESM1.2, which has a higher resolution than MPI- ESM-1-2-HAM [76,96] |
| JULES | HadGEM3-GC31-LL<br>HadGEM3-GC31-MM<br>UKESM1-0-LI | HadGEM3-GC31 is a coupled atmosphere–land–ocean–sea ice model. Compared with HadGEM3-GC31-LL, HadGEM3-GC31-MM has a higher resolution. UKESM1 takes HadGEM3-GC31 as the core of the physical model and adds the carbon and nitrogen cycle and atmospheric chemical composition to it [98,99] |
| LM3.0 | KIOST-ESM | Atmosphere–land–ocean–sea ice coupled model [68,96] |
| LM4.0 | GFDL-CM4 | A coupled ocean–atmosphere model [68,96] |
| LM4.1 | GFDL-ESM4 | A fully coupled chemistry–climate model [68,96] |

By combining the simulation results of the model for the average LAI and LAI trends in Figures 2 and 3 and the different characteristics of the models in Table 2, we found that the simulation results of the models using different land surface models were quite different on the whole; the simulation results of models using the same land surface model had overall consistency, whereas the simulation results of models using different versions of the same land surface model were different. There are many possible reasons for the large difference in the simulations of vegetation growth, such as the simplified parameterization, uncalibrated parameters, and the atmospheric forcing data that drive the model. The vegetation growth in the land surface model also subject to the simulations of other processes directly affecting vegetation growth, such as the simulation of soil temperature and moisture, surface radiation transfer, etc. Using the Community Land Model (CLM) as an example, Luo et al. [100] used the simulated data of Weather Research and Forecasting Model (WRF) to apply to the forcing data sets of the CLM model in the Tibetan Plateau, and found that there are deviations between simulated and observed surface temperatures with RMSE in the range of 2.0–4.2 °C. CLM4.0 simulated [101] lower soil temperature by −0.83 °C and higher sensible heat flux up to 60 W.m$^{-2}$, except in winter at Maqu Alpine Grassland. Xie et al. [102] found that the simulation of the winter radiation balance component and the surface energy balance component of CLM4.5 was poor, especially the simulation of the surface reflected radiation with the highest RMSE of 165.16 W.m$^{-2}$ in January, and sensible heat flux in winter had a serious deviation with the highest RMSE of 145.15 W.m$^{-2}$ in February. Song et al. [103] used CLM4.5, which underestimated soil temperature and latent heat flux in winter at the Naqu site, which indicated that the parameterization schemes of snow processes and surface albedos in CLM4.5 need to be improved. All these discrepancies in land surface simulations may lead to poor simulations of vegetation growth. Mao et al. [104] found that the GPP and LAI both had a positive correlation with precipitation and a strong negative correlation with incident shortwave radiation globally. Due to the special geography of the TP, especially the complex lower cushion surface characteristics, there is a particularity and complexity of the land–air interaction in the area, which has caused difficulties for CLM land surface simulation. How to improve and perfect the simulation performance of the CLM model on the

vegetation of the TP requires more in-depth research in the future. However, there are factors that can be improved, such as continuing to optimize the parameter schemes of simulating the temperature, precipitation, radiation flux, and the coverage of snow on the TP in CLM. Although almost all CLM models overestimated LAI and the LAI trend, there were differences in the degree of overestimation. An obvious difference is that the FIO-ESM-2-0 model with the ocean wave model added to the coupling had better performance in simulating the area-averaged LAI of the Tibetan Plateau from 1981 to 2014 (Figure 2) than other CLM models. Other modules such as the ocean wave model in the coupled model might also have had a large impact on the CLM model.

Most models had a worse performance in simulating the forest LAI and LAI trend compared with other vegetation types. The reasons for this difference may be that, compared with grasslands and meadows, the vegetation growth mechanism in forest ecosystems is more complex, the species of forest ecosystems are more abundant, and it is more difficult to establish mathematical structures for simulations with different species. Changes in the long-term processes of different species within the forest system are more complex, and it is more difficult to establish mathematical structures with simulations.

From CMIP5 to CMIP6, the average LAI over the Tibetan Plateau still showed overestimation but of an even higher magnitude. Bao et al. [37] found that 10 out of 12 CMIP5 models overestimated LAI with bias of between 0.44 and 3.6 $m^2$ $m^{-2}$ from 1986 to 2005. We found that 25 out of 35 CMIP6 overestimated LAI of TP, with bias ranging from 0.07 to 5.38 from 1981 to 2014. For the same model from CMIP5 to CMIP6, we found that some models had better performance: for example, HadGEM3-GC31 had the smallest bias of the CMIP6 models. Some models showed poor performance in CMIP6—for example, CESM2 in CMIP6 showed much a higher average LAI than its previous version, CCSM4 in CMIP5; additionally, INMCM4, with the lowest bias of 12 CMIP5 models [37], ranked 23rd in area-averaged bias among the 35 CMIP6 models. Both CanESM2 from CMIP5 and CanESM5 from CMIP6 maintained a better simulation of the average LAI on the TP with the smaller bias, the same as the MPI-ESM1-2-HR and the old version MPI-ESM-LR. There were also models, whether in the CMIP5 or in the CMIP6, where the simulation performance was relatively poor, such as bcc-csm1.1-m and the new version, BCC-CSM2-MR, in CMIP6, and NorESM1-ME and NorESM2-MM/LM from CMIP5 to CMIP6.

Song et al. [105] found that CMIP6 generally overestimated the global multiyear average LAI, and the overestimation of growing season length (GSL) contributed to the overestimated LAI in boreal and some temperate areas. We found that CLM family also overestimated the average LAI during the growing season in 1981–2014 on the TP. We analyzed the monthly average LAI of 35 models from 1981 to 2014 and found that most of the models had a longer growing season (Figure S8). CMIP6 LAI in April, October, and November were still large. Part of the reason for the global multi-year average LAI and the TP LAI overestimation was the same. Moreover, we found that LAI increased greatly during the leaf emerge stage in most CLM family models, which suggested too much carbon was being allocated to leaves. Improving the phenology and carbon allocation is crucial for improving LAI simulations over the Tibetan Plateau.

Climate change has led to changes in vegetation on the TP in recent decades. From the 1980s to the beginning of the 21st century, the vegetation coverage rate of the TP showed an overall increasing trend [21], with large seasonal and spatial variations. The spring vegetation coverage of the Tibet Plateau showed the larger increasing rate [106] than other seasons. The humid areas in the Southeast TP showed increasing vegetation coverage while the Central and Northwest TP showed declined vegetation coverage [21,107]. The upper limit of the vertical natural zone of vegetation over the TP has changed significantly. The forest lines migrated to high altitudes [107]. The glacier retreat and permafrost ablation will aggravate the degradation of regional alpine grassland [108] on the TP. Due to changes in the permafrost environment, the soil moisture and nutrients in the root layer of vegetation are decreased, resulting in the drying out of swamp wetlands and the transformation into meadows in Zoige, according to the measured data on

temperature precipitation [109], and shrub invasion of alpine meadows [107]. Species diversity in the native Kobresia humilis meadow community decreased in a simulation of a five-year temperature increase run a greenhouse in TP [110]. The degradation of permafrost, the drying out of some swamps, and the aggravation of surface salinization all exacerbated the desertification of permafrost area in the TP [111]. Meanwhile, many of the variables that cause changes in vegetation growth in the context of global change have also changed. Temperature and precipitation, which have a positive correlation with LAI [112], showed an overall increasing trend on the TP, with warming of 0.4 °C. 10 yr⁻¹ over the last 30 years [12,13] and precipitation increasing by 1.96 mm.10 yr⁻¹ in 1994–2015 [14]. Zhu et al. [113] found that in the past 50 years, the highest value of Photosynthetically Active Radiation (PAR) in China appeared in the southwest of the Tibetan Plateau (with an annual PAR of 35 mol.m⁻²d⁻¹), while the PAR in the northwest of the Tibetan Plateau showed an upward trend in different seasons. By analyzing the daily temperature data provided by the National Meteorological Information Center, China Meteorological Administration, for the Tibetan Plateau stations from 1961 to 2007, Fan et al. [114] found that spring and summer are starting earlier while autumn and winter are starting later.

Some of these changes can be monitored by remote sensing, e.g., glacier retreat [115], widespread grassland variation [116] with grassland biomass dynamics [117], rising forest lines, shrub intrusion into alpine meadows, etc. However, it is difficult for vegetation growth models to simulate these complex processes. The phenology and allocation schemes were not designed to capture tree line migration or grassland transformation. Moreover, the land surface model also could not simulate the well permafrost thawing or the glacier retreat processes over the Tibetan Plateau.

Some researchers also found that the model had large errors in other simulation variables on the TP. Xiao et al. [118] evaluated the performance of the state-of-the-art global high-resolution models in simulating hourly precipitation and extreme precipitation in summer over the TP in 1950–2050 with eight CMIP6 high-resolution models (HighResMIP) and found that the CMIP6 HighResMIP overestimated the precipitation amount and frequency. Chen et al. [119] found that, although the CMIP6 models could simulate the spatial distribution characteristics of the average annual precipitation on the Tibetan Plateau, this was generally overestimated, with an average of more than 397.8 mm.a⁻¹. The simulations of temperature and precipitation, which have a greater impact on the LAI simulation of vegetation, showed a large error in the TP. The inaccuracy of the temperature and precipitation simulation may also be one of the reasons for the large error in vegetation simulations on the TP.

The acquisition of field data in TP was limited due to geographical, topographical, and environmental factors. However, continuous actual observation data from the plateau site are also very important for the accurate description of land–atmosphere interactions and the improvement of the parameterization of different physical processes [120–122].

Therefore, there are three pathways that may improve the performance of models in simulating LAI over the TP. The first is to incorporate missing physical mechanisms that directly or indirectly impact on vegetation growth, such as aerosol effects [123], elevated CO2 concentration, and the impact of volcanic eruptions on the climate [124]. Moreover, incorporating land surface processes such as permafrost thawing processes and the winter surface parameterization scheme [102] may be particularly important over the TP. The second is to calibrate and optimize the internal parameters [104] to better represent vegetation growth over the TP. Some of the parameters were not calibrated or validated over the TP, so using artificial intelligence to train models could improve the model simulations. The third is to further improve the observation system and obtain continuous and complete atmospheric observations, as site-observed vegetation growth is also very important for improving simulations of the vegetation on the TP.

As the temperature continues to rise, the impact of the climate on plant phenology becomes more complex [125] and the acquisition of the forcing data becomes harder due

to the extreme weather problems caused by global warming, which will make simulation of the vegetation growth in the Tibetan Plateau more challenging in the future.

## 5. Conclusions

In this study, we evaluated the performance of CMIP6 models in simulating LAI and the LAI trend during the growing season of the Tibetan Plateau over the period 1981–2014, compared with the GLASS LAI. We found the following:

1.  In total, 40% of the models overestimated the greening, 48% of the models underestimated the greening, and 11% of the models showed a declining LAI trend for 1981–2014 over the Tibetan Plateau. For the LAI, 70% of the models overestimated this, while about 17% of the models underestimated it.
2.  Both the models underestimating greening, and the models underestimating LAI, showed the greatest underestimation bias in July and August. The biases and ratio of LAI (with the exception of the CLM family) and trend between the simulations and observations had the same change during the growing season.
3.  CMIP6 models overestimated the LAI trend of alpine vegetation, forest, and grassland, but underestimated the meadow and shrub. The greening of grasslands was overestimated, and the greening of meadows was underestimated in CMIP6. Compared with other vegetation types, the performance of simulating the forest LAI trend was poor with the highest RMSE, and the declining trend in forest pixels showing a declining trend on the TP, was generally underestimated.
4.  The performance in simulating the spatial distribution of LAI was better than the LAI trend. The underestimation of LAI was mainly in meadows and alpine forest areas in southeast TP. Similar to the forest LAI trend, the simulation performance of forest LAI was also poor, with the highest RMSE, and the forest LAI in parts of the southeast where alpine forests were concentrated on the TP was underestimated by 20 of 35 CMIP6 models.

**Supplementary Materials:** The following supporting information can be downloaded at: https://www.mdpi.com/article/10.3390/rs14184633/s1. Figure S1. The relative bias of the monthly mean LAI with simulations and observations; Figure S2. The bias of the monthly mean linear trend with simulations and observations; Figure S3. Spatial distributions of the simulated LAI during the growing season; Figure S4. The distribution of bias, relative bias and RMSE between the simulated and observed LAI with 35 CMIP6 models for different vegetation types; Figure S5. The area-averaged linear trend of simulated and observed LAI the during the growing season ($p < 0.05$); Figure S6. Spatial distributions of the simulated LAI linear trend during the growing season; Figure S7. The distribution of bias and RMSE between the simulated and observed LAI trend with 35 CMIP6 models for different vegetation types; Figure S8. The monthly LAI of 35 CMIP6 models from 1981 to 2014; Table S1. Summary of evaluation metrics and error ranking for models with the performance to simulate the LAI of Tibetan Plateau during growing season in1981–2014; Table S2. Summary of evaluation metrics and error ranking for models with the performance to simulate the LAI trend of Tibetan Plateau during growing season in1981–2014.

**Author Contributions:** Writing—original draft preparation, J.L. and Y.L.; writing—review and editing, J.L. and Y.L. All authors have read and agreed to the published version of the manuscript.

**Funding:** This research was funded by the Strategic Priority Research Program of Chinese Academy of Sciences grant number No. XDA20050102, XDA23060601, and the National Natural Science Foundation of China grant number No. 41975135.

**Data Availability Statement:** The study did not report any data.

**Acknowledgments:** Thank you to the anonymous reviewers who gave useful advice and made our article more organized. This study was supported by the Strategic Priority Research Program of Chinese Academy of Sciences (No. XDA20050102, XDA23060601) and the National Natural Science Foundation of China (No. 41975135). We are grateful for the data set of the Tibetan Plateau Boundary high frequency (HF) and the 1:1 million vegetation data set in China provided by the National Tibetan Plateau Data Center (http://data.tpdc.ac.cn, accessed on 9 December 2021). We thank the

Beijing Normal University for providing the Global Products of Essential Land Variables (GLASS). We acknowledge the World Climate Research Programme, which, through its Working Group on Coupled Modelling, coordinated and promoted CMIP6. We thank the climate modeling groups for producing and making available their model output, the Earth System Grid Federation (ESGF), for archiving the data and providing access, and the multiple funding agencies that supported CMIP6 and ESGF.

**Conflicts of Interest:** There is no conflict of interest.

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
