# Peer review of "How Well Do CMIP6 Models Simulate the Greening of the Tibetan Plateau?"

_remotesensing, doi:10.3390/rs14184633_

Round 1
Reviewer 1 Report
This study evaluates the accuracy of LAI of CMIP6 models in the Tibetan Plateau by comparing it with GLASS LAI. For the mean and trend of LAI during the target period of time, the spatial mean and spatial distribution in the target region are investigated. The topic is suitable for this journal, and the manuscript is generally well structured. However, overall, this manuscript appears to be more of a report on the model evaluation than a research article. The originality and new findings are not clear. Major revisions are required for publication as a research article.
General comments
1. This manuscript is more like a report describing the model accuracy than a research article. According to papers cited in the manuscript, it has been shown that CMIP5 models generally overestimate LAI in the Tibetan Plateau, and CMIP6 models overestimate global mean LAI for non-forested areas. How do the results in the present study differ quantitatively from those in the previous study? Is the bias larger or smaller? What are the causes? What are the results specific to CMIP6 models in the Tibetan Plateau? Since we already know that models are prone to overestimate LAI for non-forested areas, a tendency specific to the Tibetan Plateau and specific error factors that can lead to the model improvement in the Tibetan Plateau and surrounding regions should be discussed. The metrics and evaluation methods used in the present study seem to be similar to those used in previous studies, and it appears that the authors simply did the same thing in the Tibetan Plateau that has been done for the globe and models with different versions in previous studies.
Specific comments
2. lines 157–159: What is the spatial resolution of original simulation data? Was it enough for analysis with 0.5° x 0.5°spatial resolution?
3. line 172: What is “trend”? It seems to be the slope of a linear regression line for the time series LAI data but should be defined clearly. According to the introduction, this study focuses on the growing season. If only data during the growing season was used to calculate the trend, that should also be explained here. The growing season itself is also not defined.
4. Lines229–230: The LAI bias is not explained in section 2.4. In section 2.4, there are only explanations of metrics for LAI trend and no explanations of metrics for LAI. In Figure 2, there is no unit on the vertical axis.
5. Lines239–241: At this stage, results showing the greening trend are not presented. Therefore, it is not clear whether a ratio value greater than 1 indicates an overestimation of greening trend or that of declining trend.
6. Line253: The monthly variations of both LAI and LAI trend are not explained in section 2.4.
7. Lines255–259: Not only the monthly variation but also the LAI bias is not explained in section 2.4. Which figure shows the results described here? Figure 4? Figure 4 is not cited in the text. Figure 4 contains two graphs without any explanation (the same applies to Figure 5).
8. Lines265–273: The overestimation and underestimation described here appear to be only slight. Is it remarkable enough to explain? Are the results meaningful given the uncertainty in the model?
9. Line274: The relative LAI bias is not explained in section 2.4. Forest LAI is usually greater than grassland LAI. Therefore, it is natural that the absolute value of the bias for forest is larger than that for grassland. On the other hand, the relative bias for grassland is expected to be larger than that for forest because grassland LAI is usually very small. In the conclusion section, the authors state that simulation performances of LAI and LAI trend were worse for forest. This seems like a natural result as long as the absolute bias value is used to evaluate the performance.
10. Line284: The monthly LAI trend is not explained in section 2.4.
11. Lines314–317: The ranking for LAI is not described in section 2.4. If it is properly described in section 2.4, the first part of this sentence seems redundant and unnecessary.
12. Lines319–321: The pattern correlation is not defined for LAI in section 2.4 (only defined for LAI trend). In the figure, where is the pattern correlation value shown? Is it the number in parentheses in Figure S3? Please add explanations. Why not show the pattern correlation value in Figure 7 instead of Figure S3?
13. Lines334–335: The overestimation of GFDL-ESM4 appears to be distributed to the central to northeast area rather than to the southeast.
14. Line339: It appears that forest LAI is not so overestimated.
15. Lines341–343: The ranking of HadGEM3-GC31-LL is high, and the pattern correlation value for HadGEM3-GC31-LL seems to be large. Doesn't this model represent the LAI distribution well? What is the pattern correlation value to evaluate that the model reproduces the LAI distribution well?
16. Lines348–351: The LAI bias in forest does not appear to be so remarkable. Looking at Figure 7, the statements “forest LAI was generally poorly simulated” and “All 35 models had large bias in forest areas” do not seem to be true.
17. Lines372–374: In the figure, where is the pattern correlation value shown? Is it the number in parentheses in Figure S5? Please add explanations. Why not show the pattern correlation value in Figure 9 instead of Figure S5?
18. Lines374–378: The former five models are not the only ones showing greening in the southeast region. The decline in AWI-ESM-1-1-LR is not remarkable.
19. Lines378–379: Why is MPI-ESM-1-2-HAM the best, even though it is #2 in the ranking?
20. Lines392–395: According to Figure 9, FIO-ESM-2-0 underestimated the greening. This paragraph seems to describe the results for the remaining models; however, FIO-ESM-2-0 is included in the previous paragraph.
21. Lines399–404: According to Figure 9, I cannot understand that the simulation of forests’ LAI trend was the worst. Figure 8 shows that only a few pixels in the forest show a decreasing trend, but there is no significant trend over the forest area (white pixels are dominant). Can readers read the characteristics described here from figures? The results calculated for each vegetation type are not presented; therefore, it is difficult to understand explanations here.
22. Lines479–481: This sentence seems a bit odd. It is important to discuss the causes of the overestimation. Are the causes of the overestimation the same for the globe and the Tibetan Plateau? Is the degree of the overestimation the same for the globe and the Tibetan Plateau? Are there any characteristics specific to the Tibetan Plateau that can lead to the model improvement in and around the Tibetan Plateau?
23. Lines482–486: In Figure 7, the ranking of FIO-ESM-2-0 is not so high. The overestimation and underestimation appear to cancel each other out, which leads to the low area-averaged bias. According to Figure 9, the ranking of FIO-ESM-2-0 is also not high, and the trend is underestimated or is estimated as opposed to GLASS. How can the authors conclude that the ocean wave model is important to simulate the vegetation LAI and its variations? Even if the ranking of FIO-ESM-2-0 was high, is the ocean wave model really more important than other factors?
24. Lines497–501: The results calculated for each vegetation type are not presented; therefore, it is difficult to understand that models showed worse performance for forest. Forest LAI is usually greater than non-forest LAI. Therefore, it is natural that results for forest were worse as long as the absolute bias value is used to evaluate the performance.
25. Lines503–505: What is the basis for this inference? Isn't it possible that NPP is underestimated due to the underestimation of LAI?
26. Lines505–513: Are these sentences related to the underestimation of forest LAI? In the previous paragraph, the overestimation of precipitation is referred to as a cause of the overestimation of LAI.
27. Lines541–544: The results calculated for each vegetation type are not presented; therefore, it is difficult to understand these conclusions.
Minor Comments
28. Line86: Is the form of citation correct?
29. Line103: Even though it is not the first appearance, LAI is spelled out. It should be spelled out at the first appearance.
30. Line103: What is EMS?
31. Line104: What is ESM?
32. Line118: Even though it is not the first appearance, LAI is spelled out.
33. Lines131–133: Is the forest area in the south also part of the Tibetan Plateau? It looks like mountains surrounding the plateau.
34. Line134: The latitude of the target region is not so high.
35. Line143 (Figure 1): What data is this? Please describe the source of the data. Are the latitude and longitude in the figure correct? Please check. What year is the data from? The distribution of vegetation type seems to change during the target period of time. What are the white areas in the Tibetan Plateau? The image resolution is low and the text in the image is difficult to read (the same applies to figures 6–9).
36. Lines145–148: Which version of the GLASS data? Wouldn't it be better to describe the site where you downloaded the data?
37. Line155: Wouldn't it be better to describe the site where you downloaded the data?
38. Lines180–181: What is the average trend? Isn't one trend value calculated for each cell?
39. Line187: What is the trend mean? Isn't one trend value calculated for each cell?
40. Line188: What is TP?
41. Lines188–190: Isn’t this explanation also related to equations (1) and (3)? If the authors were to include an explanation, wouldn't it be before equation (1)? However, this explanation seems unnecessary.
42. Lines239–241: Which figure shows this result? Figure 3? Explanations are made before the figure is cited. There are other such cases through the text.
43. Line265: What is C-Earth3-Veg-LR?
44. Line276: Does the relative LAI bias have a unit?
45. Lines284–285: Which figure shows this result? Figure 5? Explanations are made before the figure is cited.
46. Line296: Does the ratio have a unit?
47. Lines330–332: For BCC-ESM1, the overestimation in grassland does not appear to be remarkable.
48. Line333: What does “GFDL-CM4 in the LM family (except for KIOST-ESM)” mean?
49. Lines380–388: Which figure corresponds to the descriptions here? Figure 9? Figure 9 is not cited in the text. UKESM1-O-LI and GFDL-ESM4 appear to capture the greening trend in the southwestern region.
50. Line414: Don’t hatched areas indicate that models simulated different trend from GLASS?
51. Lines546–547: This paragraph seems to describe the spatial distribution of LAI; however, suddenly the average LAI is mentioned. “70% of the models overestimated LAI” has already been stated in the first item of the conclusion (lines 533–536).
Reviewer 2 Report
The authors used remote sensing data, LAI data from the Global Land Surface Satellite (GLASS) dataset from the 1981–2018 period to reproduce the greening trend in simulations of CMIP6 models in the Tibetan Plateau region.
Overall, the paper covers an important and necessary topic in analyzing the performance of International Coupled Model Comparison Program models (in its sixth generation - CMIP6) to validate the trends of grassland greening in the Tibetan Plateau region. However, this paper needs to provide more critical details on the use of remote sensing data and the relevance of the validation processes for a better understanding of the climate change impacts on vegetation.
Please, see specific comments below:
Line 33: Authors should state the NDVI as the Normalized Difference Vegetation Index (NDVI).
Line 33: Please, rephrase this sentence: “The increase in temperature has led to an overall increase in NDVI”. The increase in NDVI results from a process not limited to the change in temperature, and the NDVI is a physical indicator only.
Line 36: The vegetation structure is one of the components linked to the NDVI variation, and in this manuscript, a few lines address this issue.
Line 54: Most of the review literature is based on the NDVI, not the LAI indicator.
Line 68: This is not a limitation: “they cannot study the dynamic changes in vegetation under future climate scenarios”.
Line 80-81: Authors should rephrase this sentence. Again, it is not about limitations. The variation of the accuracy of the simulation results is a limitation instead.
Line 121: Authors should compare the accuracy of the simulation results considering at least two types of remote sensing data: LAI and NDVI. The grassland vegetation structure (and other vegetation types) is not fitly captured by a single physical indicator such as the LAI. Incorporating at least the NDVI dataset into the simulations would enrich the discussion of the manuscript. In addition, the manuscript lacks a more profound discussion regarding how different types of vegetation and component structures respond to climate change and how this response is measured by remote sensing products and simulations.
Figure 1: Please, include the location of the Tibetan Plateau region in the globe.
Line 153: A single citation is insufficient to justify using the LAI as the reference database.
Line 419: LAI alone is not enough indicator.
Line 422: The Amazon forest is quite different from the vegetation types described in the Tibetan Plateau region.
Table 2: Authors should include this table in the material and methods section.
Discussion and conclusions should be more oriented to the climate change impacts on the Tibetan Plateau vegetation.
My recommendation is to reconsider after major revision. My recommendation is to incorporate at least the NDVI dataset into the simulations in order to enrich the discussion of the manuscript.
Reviewer 3 Report
This manuscript presents a comparative study on the leaf area index (LAI) simulations of 35 Earth System Models that participated in CMIP6 to a remote-sensing-derived LAI product (GLASS LAI). Generally, it is of interest for the scientific community. The paper is well written and structured. However, there are still some questions and suggestions for this study before it can be finally accepted.
1. In the last second paragraph of Introduction section, the authors should emphasize the research gap and novelty of this work.
2. Line 198, “2.4.2. significant test method” should be “2.4.2. Significant test method”.
4. Fig. 4, it is difficult to identify the symbols in b/w in print. Please change the symbols. Additionally, the subtitles of the two graphs are missing.
4. Figs. 7 and 9, the quality should be improved significantly. Axis labels, titles and legends are not clear and readable.
5. The differences between the present simulation results of various models are pretty huge. Please explain the reasons behind these significant departures.
6. Based on the present results, it is good to give some implications on how to apply the existing models for a better accuracy.
Reviewer 4 Report
The manuscript "How well do CMIP6 models simulate the greening of the Tibetan Plateau?" is an excellent effort by Liu et al in bringing to light the fidelity of CMIP6 models in simulating the greening of Tibetan plateau. Proper justification is provided in using GLASS LAI dataset and this study is timely and appropriate in the use of its methodology, results and conclusions. It may be accepted after minor edits in language and the following suggestions:
Line number 209: two full stops
Line 456: CMIP6 model -> CMIP6 models
Line 473: "Although CLM is one of the state-of-the-art land surface models, the CMIP6 CLM family showed poor simulations of both LAI and the LAI trend" - There are two aspects that the authors can add here in the conclusion or in the literature described in the introduction. One is a small commentary on how to ensure that the CLM based models be improved? I.e. do the authors think that recent advances in artificial intelligence methods or adding missing physics in the existing models is the way to go.
Second suggestion is related to the relationship of greening to photosynthetically active radiation (PAR) and somewhat related to the missing physics leading to model biases raised in the first point. PAR is related to the greening over land and for the case of China it has been shown in studies like Zhu et al 2010. Further, the authors can add on how the important physical processes such as volcanoes, aerosols and others which might be missing in the models may lead to improvement in the LAI. The studies such as Swingedouw et al 2017, Singh et al 2020 and others show the importance of volcanic processes in improving the climate models. There are other coupled processes which may help in improving the LAI which can be discussed.
Zhu, X., He, H., Liu, M., Yu, G., Sun, X. and Gao, Y., 2010. Spatio-temporal variation of photosynthetically active radiation in China in recent 50 years. Journal of Geographical Sciences, 20(6), pp.803-817.
Swingedouw, D., Mignot, J., Ortega, P., Khodri, M., Menegoz, M., Cassou, C. and Hanquiez, V., 2017. Impact of explosive volcanic eruptions on the main climate variability modes. Global and Planetary Change, 150, pp.24-45.
Singh, M., Krishnan, R., Goswami, B., Choudhury, A.D., Swapna, P., Vellore, R., Prajeesh, A.G., Sandeep, N., Venkataraman, C., Donner, R.V. and Marwan, N., 2020. Fingerprint of volcanic forcing on the ENSO–Indian monsoon coupling. Science advances, 6(38), p.eaba8164.
Round 2
Reviewer 1 Report
The manuscript has been improved significantly. However, there are still many unclear points and notational errors. The following points should be clarified before publication.
1. Section 2.4.1: The definition of average LAI seems to vary in the text. The “average LAI” in lines 202 and 203 appears to indicate the average growing season LAI for each year. The “average LAI” in lines 212, 222, 233, and 245 seems to indicate the average growing season LAI from 1981 to 2014. Moreover, it is not clear whether it is an area average or a value for each grid cell. For example, “average LAI” in line 204 seems to imply the area average. The text should be revised to more clearly describe the actual calculation.
2. Lines 249–254: The sentences “And, the ratio …but with the overestimation.” are confusing. They do not seem to correspond to equation (6). Is it correct that the sentences are here?
3. Section 3.1: At the time of the previous manuscript, I understood the meaning of the ratio of trend (<0 indicates that trend was not captured, etc., as the authors added explanations on lines 249–254 of the revised manuscript). My concern is that readers cannot know whether a ratio of trend > 1 indicates an overestimation of the greening trend or overestimation of the declining trend. If the area average trend of GLASS shows greening, it should be explained, which is supposed to help the reader understand this paragraph.
4. Lines 419–425: Figure S4 clearly shows results for each vegetation type, and I understand that the authors state that performance for forest LAI was poor based on RMSE. According to Figure S4, the median value of absolute LAI bias for the forest is almost zero and does not show a clear overestimation or underestimation trend. According to Figure 7, only about 20 models, not most models, show clear underestimation in areas where forests are concentrated on the southern edge of the TP. Therefore, some of conclusion 4 (Lines 772–777) do not appear to be supported by the results.
5. Lines 458–462: Please revise the text according to your answer to my comment 18 on the previous manuscript.
6. Lines 468: What is “overestimation and underestimation of polarization”?
7. Lines 490–492: Figure 8 shows that only a few forest pixels show a declining trend. According to Figure 9, most models do not show a ratio value between 0 and 1 (underestimation of the trend) for the forest pixels with a declining trend. Therefore, the description here (the declining trend … concentrated (Figure 9).) and that in conclusion 3 (the declining trend … generally underestimated.) are not readily apparent from the figures.
8. Errors in the notation of units (the followings are some examples. The authors should check throughout the manuscript)
Table 1: 250KM → 250 km
Line 379: There must be a space between m2 and m-2.
Line 651: A period is not required between m2 and m-2.
Line 651: Numbers representing powers should be superscripted.
Line 704: Unlike other places, a slash is used.
9. Typos (the followings are some examples. The authors should check throughout the manuscript)
Line 122: bias → difference?
Line 183: Many → many
Line 659: .Both→ . Both
Line 667: while → While
Figure S7(b): Is the unit of the vertical axis correct?
Abbreviation: Even for a word that appeared in the abstract, it should be spelled out at the first appearance in the main text.
The authors have revised the manuscript substantially. The revised manuscript should be proofread by a proficient English speaker.
Reviewer 2 Report
Accept in present form.
Author Response
Dear reviewer: Thanks for your decision. And your constructive comments are very important on my manuscript.Reviewer 3 Report
The authors have satisfactorily addressed the concerns of reviewer.
Author Response
Dear reviewer: Thanks for your comments. And your constructive comments are very important on my manuscript.